# <sup>14</sup>C-based separation of fossil and non-fossil CO<sub>2</sub> fluxes in cities using relaxed eddy accumulation: results from tall-tower measurements in Zurich, Paris, and Munich

Ann-Kristin Kunz<sup>1, 14</sup>, Samuel Hammer<sup>2,3</sup>, Patrick Aigner<sup>4</sup>, Laura Bignotti<sup>5</sup>, Lars Borchardt<sup>6</sup>, Jia Chen<sup>4</sup>, Julian Della Coletta<sup>2,3</sup>, Lukas Emmenegger<sup>7</sup>, Markus Eritt<sup>6</sup>, Xochilt Gutiérrez<sup>6</sup>, Josh Hashemi<sup>9</sup>, Rainer Hilland<sup>1</sup>, Christopher Holst<sup>10</sup>, Armin Jordan<sup>6</sup>, Natascha Kljun<sup>8</sup>, Richard Kneißl<sup>6</sup>, Changxing Lan<sup>10</sup>, Virgile Legendre<sup>6</sup>, Ingeborg Levin<sup>2,†</sup>, Benjamin Loubet<sup>5</sup>, Matthias Mauder<sup>11</sup>, Betty Molinier<sup>8</sup>, Susanne Preunkert<sup>2,3</sup>, Michel Ramonet<sup>12</sup>, Stavros Stagakis<sup>13</sup>, and Andreas Christen<sup>1</sup>

Correspondence: Ann-Kristin Kunz (ann-kristin.kunz@meteo.uni-freiburg.de)

Abstract. Relaxed eddy accumulation (REA) measurements for <sup>14</sup>CO<sub>2</sub> enable the estimation of fossil fuel (ff) CO<sub>2</sub> fluxes in urban areas. This work is based on 252 REA ffCO<sub>2</sub> flux measurements conducted on tall towers in the cities of Zurich, Paris, and Munich. The ffCO<sub>2</sub> fluxes were compared to net eddy covariance CO<sub>2</sub> fluxes to quantify the role of non-fossil (nf) CO<sub>2</sub> fluxes. In all three cities, winter CO<sub>2</sub> fluxes were predominantly fossil, with mean ffCO<sub>2</sub> contributions of about 80%. Summer fluxes could be most clearly partitioned in Munich, where improvements in the REA setup, the <sup>14</sup>CO<sub>2</sub> measurement precision, the sampling strategy, and the source strength increased the signal-to-noise ratios compared to Zurich and Paris. In Munich, the observed nfCO<sub>2</sub> fluxes were predominantly positive (~50% of net summer fluxes), demonstrating the major role of respiration, biofuels, and certain industrial processes. Particularly large nfCO<sub>2</sub> fluxes from the direction of a brewery suggest non-respiratory anthropogenic contributions and highlight the complexity of urban environments. Additionally, the absolute CO<sub>2</sub> and <sup>14</sup>CO<sub>2</sub> concentrations of the REA samples were compared to clean background concentrations to estimate

ffCO<sub>2</sub> excess concentrations. Across all cities, ffCO<sub>2</sub> contributions to regional excess concentrations were much lower (< 65%

<sup>&</sup>lt;sup>1</sup>Chair of Environmental Meteorology, Faculty of Environment and Natural Resources, University of Freiburg, Freiburg, Germany

<sup>&</sup>lt;sup>2</sup>Institute of Environmental Physics, Heidelberg University, Heidelberg, Germany

<sup>&</sup>lt;sup>3</sup>ICOS Central Radiocarbon Laboratory, Heidelberg University, Heidelberg, Germany

<sup>&</sup>lt;sup>4</sup>Professorship of Environmental Sensing and Modeling, Technical University of Munich, Munich, Germany

<sup>&</sup>lt;sup>5</sup>ECOSYS, INRAE, AgroParisTech, Université Paris-Saclay, Palaiseau, France

<sup>&</sup>lt;sup>6</sup>ICOS Flask and Calibration Laboratory, Max Planck Institute for Biogeochemistry, Jena, Germany

<sup>&</sup>lt;sup>7</sup>Empa, Materials, Science and Technology, Dübendorf, Switzerland

<sup>&</sup>lt;sup>8</sup>Centre for Environmental and Climate Science, Lund University, Lund, Sweden

<sup>&</sup>lt;sup>9</sup>Alfred Wegener Institute, Helmholtz Centre for Polar and Marine Research, Potsdam, Germany

<sup>&</sup>lt;sup>10</sup>Institute of Meteorology and Climate Research, Atmospheric Environmental Research, Karlsruhe Institute of Technology, Garmisch-Partenkirchen, Germany

<sup>&</sup>lt;sup>11</sup>Institute of Hydrology and Meteorology, Dresden University of Technology, TUD, Tharandt, Germany

<sup>&</sup>lt;sup>12</sup>Laboratoire des Sciences du Climat et de l'Environnement, CEA, CNRS, Université Paris-Saclay, Gif-sur-Yvette, France

<sup>&</sup>lt;sup>13</sup>Department of Environmental Sciences, University of Basel, Basel, Switzerland

<sup>&</sup>lt;sup>14</sup>Previously at: Institute of Environmental Physics, Heidelberg University, Heidelberg, Germany

<sup>†</sup>deceased, 10 February 2024

in winter and < 30% in summer) than to local eddy covariance  $CO_2$  fluxes, demonstrating fundamental differences between local and regional  $CO_2$  fluxes. The combination of  $^{14}CO_2$  observations and the REA method is a sophisticated approach that challenges the limits of current analytical capabilities, while providing unique opportunities for quantifying ff $CO_2$  and nf $CO_2$  fluxes.

#### 1 Introduction

15

Cities are hotspots for fossil fuel (ff) CO<sub>2</sub> emissions and are at the heart of emission reduction efforts. To guide and monitor the pathways of cities towards climate neutrality, measuring and modeling urban ffCO2 emissions is essential. While total CO2 fluxes can be measured using the eddy covariance (EC) method, direct observations of fossil or non-fossil CO2 are lacking. However, a separation of the two components is important because, in addition to ffCO<sub>2</sub> emissions, biospheric and human respiration fluxes play a substantial role in the urban carbon budget (e.g. Kellett et al., 2013; Miller et al., 2020; Wu et al., 2022; Stagakis et al., 2025). Several studies have attempted to separate ffCO<sub>2</sub> and nfCO<sub>2</sub> fluxes. Wu et al. (2022) combined CO<sub>2</sub> fluxes from EC measurements and CO fluxes from flux-gradient measurements to estimate turbulent ffCO<sub>2</sub> fluxes on a tower in Indianapolis 30 m above ground level, assuming a constant CO/ffCO2 flux ratio. The latter was determined from CO and <sup>14</sup>CO<sub>2</sub> concentration measurements of flask samples collected weekly at the measurement site and an upwind background station, following Levin et al. (2003). Hilland et al. (2025) proposed a linear mixing model to separate biospheric, road traffic, and stationary combustion CO2 fluxes using simultaneous tall-tower EC measurements of CO2 and co-emitted species (CO and  $NO_x$ ), as well as sector-specific, constant flux ratios determined from a bottom-up emission inventory. Other studies used <sup>14</sup>CO<sub>2</sub> observations to separate fossil and non-fossil CO<sub>2</sub> enhancements relative to a background concentration (e.g., Levin et al., 2003; Turnbull et al., 2015; Miller et al., 2020). In this case, surface emissions can be estimated using atmospheric transport models or the Radon-Tracer-Method, for example (Levin et al., 2003; Maier et al., 2024b). The source area thereby depends on the choice of the background station and includes a large region beyond the city boundaries if a tropospheric or continental clean air background site is used (Turnbull et al., 2015). To our knowledge, all previous studies estimating urban ffCO<sub>2</sub> emissions relied on bottom-up information, inverse modeling results, or assumed constant proxy/ffCO<sub>2</sub> ratios, despite the fact that ratios such as CO/ffCO2 vary significantly with fuel carbon content and combustion conditions (Turnbull et al., 2015; Maier et al., 2024a).

We overcome these limitations using  $^{14}\mathrm{CO}_2$  relaxed eddy accumulation (REA) measurements, as first described in Kunz et al. (2025). On a tall tower over the city, air is conditionally collected for one hour in an updraft or downdraft reservoir using fast-switching sampling valves that respond to a 20 Hz vertical wind signal from a 3D ultrasonic anemometer. Transfer of the collected air to portable glass flasks enables  $^{14}\mathrm{CO}_2$  and  $\mathrm{CO}_2$  measurements in a subsequent laboratory analysis, and thus the estimation of  $\mathrm{ffCO}_2$  concentration differences between updraft and downdraft samples. Combined with net  $\mathrm{CO}_2$  fluxes measured by open-path or closed-path EC, this novel approach enables the estimation of  $\mathrm{ffCO}_2$  fluxes for the respective, hour-long sampling periods.

In Kunz et al. (2025), the REA flask sampling system was described and its performance was analyzed in detail. It was shown to meet high technical requirements, e.g., fast and accurate switching between updraft and downdraft sampling, while maintaining a constant flow rate in sampling and non-sampling modes. For the estimation of ffCO<sub>2</sub> fluxes, uncertainties due to the sampling procedure were negligible compared to the analytical <sup>14</sup>CO<sub>2</sub> uncertainty in the lab. Analysis of concentration differences between updraft and downdraft flask samples collected during a pilot application at a tall tower in Zurich, Switzerland, showed that separation of fossil and non-fossil components of the CO<sub>2</sub> concentration differences is feasible, but often limited by a low signal-to-noise ratio of the <sup>14</sup>CO<sub>2</sub> difference. Since then, the REA system has been further improved and operated on two tall towers in Paris, France, and Munich, Germany, for another 9 months each.

This paper presents and analyzes the ff $CO_2$  fluxes obtained from a total of 252 discrete hour-long  $^{14}CO_2$  REA measurements conducted on three tall EC towers in Zurich, Paris, and Munich. After a brief presentation of the methods (Sect. 2) and the measurement campaigns (Sect. 3), the following questions are addressed:

- 1. To what extent do <sup>14</sup>CO<sub>2</sub> REA measurements enable the separation of local fossil and non-fossil CO<sub>2</sub> fluxes in an urban area? (Sect. 4.1, 4.2, 4.3)
  - 2. (a) What are typical ff $CO_2$  and nf $CO_2$  flux contributions? (Sect. 4.3)
    - (b) Do we find indications for localized fossil and non-fossil  $CO_2$  sinks and sources, and/or observe systematic spatial and temporal differences within and between the three cities? (Sect. 4.3, 4.4)
- 3. How does the composition of surface fluxes in the vicinity of the tall tower compare to the composition of regional CO<sub>2</sub> concentration enhancements? (Sect. 4.5)

#### 2 Methods

This study analyzes the contributions of fossil and non-fossil sinks and sources to net  $CO_2$  fluxes measured successively on three different urban tall towers for about nine months each. While the net  $CO_2$  fluxes were measured continuously by the well-established EC method (e.g., Aubinet et al., 2012b), the partitioning of individual, hour-long measurements is based on REA measurements for  $^{14}CO_2$  (Kunz et al., 2025).

#### 2.1 Net CO<sub>2</sub> fluxes from eddy covariance measurements

Net turbulent CO<sub>2</sub> fluxes were computed from high-frequency CO<sub>2</sub> measurements of a closed-path (MGA<sup>7</sup>, MIRO Analytical AG, Wallisellen, Switzerland) and an open-path gas analyzer with a co-located 3D sonic anemometer (IRGASON, Campbell Scientific, Inc., Logan, UT, USA). The 20 Hz CO<sub>2</sub> measurements of the IRGASON were despiked by discarding measurements where the median absolute deviation of less than three consecutive observations was outside the upper and lower limits defined by Mauder et al. (2013). The 10 Hz measurements of the MGA<sup>7</sup> were upsampled to 20 Hz and synchronized with the IRGASON data by finding the time lag of maximum correlation between the high-frequency CO<sub>2</sub> time series. For periods with low IRGASON signal strength and consequently poor correlation between the CO<sub>2</sub> time series (correlation coefficient 

0.5), the time lag was determined via linear interpolation to correct clock drift and subsequent maximization of the covariance between  $CO_2$  and vertical wind velocity using a search window of  $\pm 0.5$  s (Hilland et al., 2025). The fluxes were then computed using the software EddyPro (Version 7.0.9, Licor Inc., Lincoln, NE, USA) with a 30 min averaging period, coordinate rotation via double rotation (Wilczak et al., 2001), and detrending via block average (Rebmann et al., 2012). High-pass filtering effects were corrected according to Moncrieff et al. (1997). For low-pass filtering effects, the correction by Moncrieff et al. (2004) was used for the IRGASON and the correction by Fratini et al. (2012) for the MGA<sup>7</sup>. Random errors of the turbulent flux estimates were calculated after Finkelstein and Sims (2001), and storage fluxes were estimated from concentrations and based on a single-point profile. Quality control flags of 0 (high quality), 1 (intermediate quality) or 2 (poor quality) were assigned to all flux estimates according to Mauder and Foken (2004), checking the assumptions of stationarity and well-developed turbulence. In addition, EddyPro outputs a large set of variables for each 30 min averaging period, including friction velocity  $u_*$ , standard deviation of vertical wind velocity  $\sigma_w$ , and molar volume of ambient air  $v_a$ . Details on the EddyPro outputs in general and the processing of the IRGASON and MGA<sup>7</sup> data in particular can be found in LI-COR (2021) and Hilland et al. (2025), respectively.

To estimate the mean CO<sub>2</sub> fluxes during the specific, typically 60 min long REA flask sampling periods (Sect. 2.2), the 30 min EC fluxes were averaged, weighted by the fraction of the EC averaging period during which REA samples were collected. This means that each 60 min flux includes two to three 30 min fluxes (usually two, since most REA measurements were scheduled at the hour). The uncertainty of the 60 min flux was estimated by error propagation of the respective 30 min random uncertainty estimates. For quality control purposes, the maximum of the 30 min quality control flags, denoted QC in the following, was considered. Since the CO<sub>2</sub> concentration measurements of the MGA<sup>7</sup> showed a better agreement with the measured flask concentrations than the IRGASON measurements (Sect. 4.1), the fluxes calculated from the MGA<sup>7</sup> measurements were used when available, otherwise the fluxes calculated from the IRGASON were used. Information on which EC data set was used is provided for each REA measurement at https://doi.org/10.5281/zenodo.17183699.

#### 2.2 14C-based separation of fossil and non-fossil CO<sub>2</sub> fluxes from relaxed eddy accumulation measurements

#### 2.2.1 REA sampling and flux calculation

100

Fossil and non-fossil components of the  $CO_2$  flux measurements were separated by  $^{14}CO_2$  analysis of flask sample pairs conditionally collected using the REA flask sampling system described in detail in Kunz et al. (2025). In summary, depending on the 20 Hz vertical wind measurements of the IRGASON's 3D ultrasonic anemometer (Sect. 2.1), air was collected through two co-located inlets with two fast-response valves into two separate reservoirs: one for updrafts, and one for downdrafts. After a sampling period of, e.g., 60 min, it was checked whether sufficient air has accumulated for a subsequent  $CO_2$  and  $^{14}CO_2$  analysis in the laboratory. If so, the accumulated air was transferred by an extended automated 24-port flask sampler into two 3 l glass flasks that could be analyzed in the laboratory (denoted as "successful" REA measurement in the following). Updraft and downdraft were thereby defined with respect to the mean vertical wind velocity  $\overline{w}$ , excluding a range of wind speeds centered around  $\overline{w}$  and scaled by the standard deviation of the vertical wind  $\sigma_w$  (scaling factor  $\delta$ ). This so-called deadband


with half-width  $\delta \cdot \sigma_w$  was intended to increase the concentration difference and to reduce the number of valve switchings (Rinne et al., 2021).  $\overline{w}$  and  $\sigma_w$  were either calculated from the 30 min period before sampling start (pre-set deadband) or dynamically adjusted using a 15 min backward-looking averaging interval (dynamic deadband). The latter lead to a more equally distributed sampling of updrafts and downdrafts and was therefore better suited for changes in vertical wind statistics during the sampling period.

Due to the costs and logistics associated with flask sample analysis, only a limited number of successful REA measurements could be analyzed. The selected REA flask samples were analyzed for  $CO_2$  and  $^{14}CO_2$  in the ICOS (Integrated Carbon Observation System) Flask and Calibration Laboratory in Jena, Germany, and in the ICOS Central Radiocarbon Laboratory in Heidelberg, Germany. Based on these measurements, the ff $CO_2$  differences between updraft and downdraft samples, in the following denoted as  $\Delta ffCO_2$ , were estimated (Appendix A1, Kunz et al. (2025)). Under stationary and well-developed turbulence, the ff $CO_2$  flux  $F_{ffCO_2}$  can then be estimated according to Eq. (1):

$$F_{\text{ffCO}_2} = \beta \,\sigma_w \,\overline{\rho_m} \,\Delta \text{ffCO}_2. \tag{1}$$

is the mean molar air density in  $\text{mol}\,\text{m}^{-3}$ . For  $\delta=0$ , the proportionality factor  $\beta$  is 0.627 if the vertical wind velocity w is normally distributed and the regression of the scalar concentration on w is linear (Wyngaard, 1992; Baker et al., 1992). For individual sampling periods, however,  $\beta$  can vary significantly, depending on the joint probability distribution of variations of the vertical wind velocity and the gas concentration (Milne et al., 1999; Fotiadi et al., 2005b; Ruppert et al., 2006; Grönholm et al., 2008). In addition,  $\beta$  decreases with increasing deadband width (e.g., Pattey et al., 1993; Fotiadi et al., 2005b). Values < 0.1 or > 1 indicate non-ideal sampling conditions for REA measurements (Grönholm et al., 2008; Hensen et al., 2009; Osterwalder et al., 2016). Due to the availability of co-located high-frequency EC measurements of net  $\text{CO}_2$  (Sect. 2.1), the measured  $\text{CO}_2$  fluxes were used to calculate  $\beta$  for each sampling period individually:

$$\beta = \frac{F_{\rm CO_2}}{\sigma_w \overline{\rho_m} \Delta \rm CO_2}.$$
 (2)

 $\Delta \mathrm{CO}_2$  is the  $\mathrm{CO}_2$  concentration difference between updraft and downdraft flask samples measured in the laboratory and  $F_{\mathrm{CO}_2}$  is the net  $\mathrm{CO}_2$  flux measured by EC (Sect. 2.1). Assuming scalar similarity between  $\mathrm{CO}_2$  and  $\mathrm{^{14}CO}_2$ , Eq. (2) can be inserted into Eq. (1):

$$F_{\text{ffCO}_2} = \frac{F_{\text{CO}_2}}{\Delta \text{CO}_2} \cdot \Delta \text{ffCO}_2 = \frac{\Delta \text{ffCO}_2}{\Delta \text{CO}_2} \cdot F_{\text{CO}_2}. \tag{3}$$

Accordingly, the fossil contribution to the net  $CO_2$  flux equals the  $\Delta ffCO_2/\Delta CO_2$  ratio of the REA flask samples. The uncertainty of the ffCO<sub>2</sub> flux was derived according to Gauss' law of error propagation from Eq. (3). For  $\Delta CO_2$  and  $\Delta ffCO_2$ , only the measurement uncertainties from the laboratory analysis were considered, as uncertainties due to the sampling process, e.g., a time lag between a change in vertical wind and a switching of the fast-response sampling valves, are negligible compared to the  $^{14}CO_2$  measurement uncertainty (Kunz et al., 2025). The uncertainty of  $F_{CO_2}$  was estimated using the random uncertainty estimate from EddyPro (Sect. 2.1).

It is important to note that Eq. (3) describes the turbulent fluxes at the measurement height. These fluxes only represent the surface fluxes if changes in the storage below the measurement height are negligible and there is no mean vertical advection.





While this is usually the case during well-mixed, convective conditions (i.e., in the afternoon), significant storage fluxes can occur, particularly in the morning hours during the transition from low-turbulence, nighttime conditions to well-developed turbulence, when the depth of the atmospheric boundary layer increases and built-up CO2 is vented from the layer below the measurement height (e.g., Stull, 1988; Crawford and Christen, 2014). A storage correction, as it is recommended and commonly applied in EC measurements (e.g., Aubinet et al., 2012b; Crawford and Christen, 2014), would require knowledge of both the storage flux  $F_{\rm CO_2,strg}$  and the ff $\rm CO_2/CO_2$  ratio of the storage fluxes. However, the magnitude of the storage flux in cities, especially in the morning, is associated with significant uncertainties (Crawford and Christen, 2014). The ffCO<sub>2</sub> contribution to the storage fluxes equals the ratio of the flux averages over the period during which CO2 accumulated below the measurement height, which does not necessarily equal the surface flux ratio during the measurement period. Consequently, a meaningful, observation-based storage flux correction for the REA measurements is not feasible. Thus, the presented fluxes are not corrected for changes in storage. While REA measurements during or after low-turbulence conditions therefore do not reflect the surface fluxes during the sampling period, the measured ffCO<sub>2</sub>/CO<sub>2</sub> ratio still provides information about the average relative contribution of fossil fuel emissions in the time period since the layer below the measurement height became decoupled prior to the start of the REA measurement – usually a nocturnal accumulation under low-wind conditions. Therefore, measurements with low turbulence and/or storage fluxes are analyzed separately. The criterion used in this study to flag the corresponding measurements is described in Sect. 2.2.4.

#### 2.2.2 REA system improvements

As the  $^{14}\text{CO}_2$  differences between updraft and downdraft samples collected in Zurich and Paris were often close to or smaller than the detection limit in the laboratory analysis, the REA system was modified, as suggested in Kunz et al. (2025). To enable the use of a larger deadband width, larger pumps were installed in the REA system before the campaign in Munich. This was necessary because a larger deadband width reduces the proportion of time during which air is collected and therefore increases the required sampling flow rate needed to collect enough air for laboratory analysis. In addition, the option for hyperbolic relaxed eddy accumulation (HREA, Bowling et al. (1999)) was added. In HREA, air is only collected if both vertical wind velocity fluctuations  $w' = w - \bar{w}$  and fluctuations in the scalar concentration  $c' = c - \bar{c}$  are above a certain threshold, which is characterized by the hole size H (similar to  $\delta$  and a pre-set or dynamic deadband in normal REA). This maximizes the concentration differences between updraft and downdraft reservoirs, as only the eddies that contribute the most to the vertical flux are sampled, and is recommended for REA applications where sampling differences are close to the detection limit (Vogl et al., 2021).

#### 2.2.3 Quality control of the REA system

To ensure high quality measurement data, the performance of the REA flask sampling system was tested regularly (for details, see Kunz et al. (2025)). To examine biases between updraft and downdraft sampling, a pair of quality control flasks was sampled about once a month by continuously collecting air through both updraft and downdraft lines without switching the valves. Simultaneously, a third flask was sampled through a separate line directly into the flask sampler, bypassing the reservoirs where







updrafts and downdrafts accumulate. If the system was operating as intended, the concentrations of the three quality control samples should agree within the WMO compatibility goal of 0.1 ppm for CO<sub>2</sub> (WMO recommendation for compatibility of measurements of greenhouse gases and related tracers (Tans and Zellweger, 2014)).

To verify the correct switching between updraft sampling, downdraft sampling, and no sampling, the measured CO<sub>2</sub> concentration differences between the updraft and downdraft REA flask pairs were compared to the CO<sub>2</sub> in situ measurements of the IRGASON and the MGA<sup>7</sup>. For this purpose, the high-frequency gas densities were converted to dry molar fractions and averaged over the respective actual sampling times, as described in Kunz et al. (2025).

To detect technical problems as early as possible, automated leak and critical component tests were carried out daily in the Paris and Munich campaigns. The results of the quality control flask measurements are given in Appendix D.

#### 2.2.4 Flagging of analyzed REA measurements

Besides technical requirements, REA is like any other turbulent flux measurement technique restricted to certain micrometeorological conditions, e.g., stationarity and well-developed turbulence (Rinne et al., 2021). Moreover, and in contrast to the EC technique, REA measurements cannot be processed retrospectively, e.g., cannot be corrected for changes in the mean vertical wind velocity. Therefore, additional criteria are necessary (Fotiadi et al., 2005a). Several criteria have already been considered in the selection of suitable flask samples during the campaigns (Kunz et al., 2025). However, due to the limited number of good sampling conditions in the urban environment, the refinement of EC processing options, and the addition of further criteria after the measurement campaign, the analyzed sampling periods were not always ideal for REA measurements. For analysis of the results, the measurements were characterized based on five flagging criteria (Table 1, see Appendix B for details). The assumptions of stationarity and well-developed turbulence were validated based on the maximum of the 30 min EC quality control flags according to Mauder and Foken (2004) (QC). Following Hensen et al. (2009) and Osterwalder et al. (2016), measurements for which Eq. (3) does not provide reasonable ff $CO_2$  fluxes were flagged according to  $\beta$  (Eq. 2). Measurements with large uncertainties due to the limited resolution of the <sup>14</sup>CO<sub>2</sub> differences between updraft and downdraft samples were flagged based on the signal-to-noise ratio SNR, defined as the minimum of the relative  $F_{\rm ffCO_2}$  and  $F_{\rm nfCO_2}$  uncertainties. Measurements during which a decoupling between the measurement height and the surface was likely were flagged using the minimum 30 min friction velocity,  $u_*$ , and the maximum of the absolute 30 min storage flux,  $|F_{\rm CO_2,strg}|$ . Based on these criteria, the analyzed REA measurements were classified into four categories (Table 1). "Well-mixed" measurements are assumed to best represent the surface fluxes during the sampling period. These measurements are the most valuable for answering our research questions and were analyzed in the most detail. In contrast, "low-turbulence" and "storage" measurements are probably not representative of the surface fluxes during the sampling period due to insufficient turbulence or changes in storage below the measurement height (Sect. 2.2.1). However, the relative ffCO<sub>2</sub> contributions were investigated to characterize the integrated fluxes before and during the sampling period. Note that the low-turbulence and storage flags are not mutually exclusive, but are closely related. Therefore, measurements with either flag were examined together. Measurements with QC = 2,  $\beta$  < 0.1,  $\beta$  > 1 or SNR 

**Table 1.** Flagging of the analyzed REA measurements based on the maximum quality control flag for stationarity and well-developed turbulence, (QC), the  $\beta$  coefficient, the signal-to-noise ratio (SNR), the minimum friction velocity  $(u_*)$ , and the maximum absolute storage flux ( $|F_{CO_2,strg}|$ ). Measurements were not considered further if any of the criteria was met ("I"); the other flags were only assigned if all criteria were met ("I").

| Flag           | QC<br>[-] |   | β<br>[-]    |   | SNR<br>[%] |   | $u_*$ [m s <sup>-1</sup> ] |   | $ F_{\rm CO_2, strg} $ [ $\mu$ mol m <sup>-2</sup> s <sup>-1</sup> ] |
|----------------|-----------|---|-------------|---|------------|---|----------------------------|---|----------------------------------------------------------------------|
| Well-mixed     | 0, 1      | & | [0.1, 1]    | & | ≥ 100      | & | $\geq 0.2$                 | & | ≤ 20                                                                 |
| Low-turbulence | 0, 1      | & | [0.1, 1]    | & | $\geq 100$ | & | < 0.2                      |   |                                                                      |
| Storage        | 0, 1      | & | [0.1, 1]    | & | $\geq 100$ |   |                            | & | > 20                                                                 |
| Not considered | 2         | / | < 0.1 / > 1 | / | < 100      |   |                            |   |                                                                      |

#### 3 Measurement campaigns





To assess the performance and to analyze the results of REA <sup>14</sup>C measurements for different urban environments, the REA system as well as the EC systems (IRGASON and MGA7) were successively installed and operated for nine months each on three tall towers in the cities of Zurich, Paris, and Munich. The measurements were conducted as part of the ICOS Cities project (https://www.icos-cp.eu/projects/icos-cities), at the same time and place as the studies by Lan et al. (2024), Stagakis et al. (2025), and Hilland et al. (2025). At each site, the gas inlets for updraft and downdraft sampling and the inlet for the MGA<sup>7</sup> measurements were mounted on a mast on top of a high-rise building or tower about 20 cm apart from the ultrasonic anemometer and the open-path CO<sub>2</sub> sensor of the IRGASON (Appendix C). The data logger, flask sampler, and the MGA<sup>7</sup> were located in a climate controlled room. Time lags due to the travel time of the sampled air from the inlets to the MGA<sup>7</sup> and the flask sampler were taken into account by synchronizing the CO<sub>2</sub> time series of the MGA<sup>7</sup> with the IRGASON (Sect. 2.1) and a site-specific rinse time, respectively (Kunz et al., 2025). REA samples were typically collected over 60 min, starting every other hour. Since increased stability at night is unfavorable for REA measurements (Fotiadi et al., 2005a), flasks were sampled during the day only. To ensure reliable measurements from the open-path gas analyzer, samples collected during periods of low signal strength, i.e., rain events, were discarded. With growing experimental experience, the logger program, REA system, and selection criteria were progressively updated, while the overall methodology remained consistent across the three cities. A documentation and version history of the logger program is publicly available at https://doi.org/10.5281/zenodo.13926681. Figure 1 shows the locations of the three measurement sites, along with the 10 - 80 % source areas for the well-mixed REA measurements. The flux footprints were derived at a 30 min resolution using the flux footprint model of Kljun et al. (2015). Table 2 provides an overview of the site-specific data. For better readability, we refer to the three sites by their respective city names.

**Table 2.** Site-specific data from the three REA measurement campaigns in Zurich, Paris, and Munich.  $\delta$  and H are the scaling factors for the deadband width (REA) and the hole size (HREA), respectively. "Pre-set" and "dynamic" indicate whether the latter was fixed at the beginning of the sampling period or continuously adjusted based on the standard deviation of the vertical wind velocity. "Successful" measurements refer to measurements in which enough sample air for laboratory analysis was collected in both the updraft and downdraft reservoirs. Finally, the numbers of successful samples selected and analyzed for  $^{14}$ C are given.

|                                           | Zurich                                 | Paris                            | Munich                            |  |
|-------------------------------------------|----------------------------------------|----------------------------------|-----------------------------------|--|
| ICOS Station ID                           | CH - Har                               | FR - Rmv                         | DE - Opd                          |  |
| Location                                  | 47° 22' 52" N<br>8° 30' 26" E          | 48° 53' 7.6" N<br>2° 25' 20.8" E | 48° 8' 50.9" N<br>11° 32' 59.3" E |  |
| Measurement height [m a.g.l.]             | 112                                    | 103                              | 85                                |  |
| Measurement period                        | July 2022 - April 2023                 | July 2023 - April 2024           | July 2024 - April 2025            |  |
| Wind directions with flow distortion [°N] | 70 - 100                               | 70 - 120                         | 340 - 20                          |  |
| Length of REA intake lines [m]            | $33 \pm 2$                             | $27\pm2$                         | 100                               |  |
| Inner diameter of REA intake lines [mm]   | 5.7                                    | 9.5                              | 8                                 |  |
| Deadband settings:                        |                                        |                                  |                                   |  |
| Regular                                   | Dynamic, $\delta = 0.7$                | Dynamic, $\delta = 0.7, 0.9$     | Dynamic, $\delta = 1.1$           |  |
| Test                                      | Pre-set, $\delta = 0.3, 0.4, 0.7, 0.8$ | Dynamic, $H = 0.6$               | Dynamic, $H = 0.8$                |  |
| Number of REA measurements:               |                                        |                                  |                                   |  |
| Started                                   | 709                                    | 498                              | 601                               |  |
| Successful                                | 338 (48 %)                             | 384 (77 %)                       | 485 (81 %)                        |  |
| With $^{14}\mathrm{C}$ and EC data        | 87                                     | 65                               | 99                                |  |

**Figure 1.** Locations of the measurement sites in Zurich, Paris, and Munich, and aggregated flux footprints of the well-mixed REA measurements according to Kljun et al. (2015) (black contour lines). The depicted areas contributed an average of 10 - 80 % to the fluxes observed during REA measurements under well-mixed conditions. Map data from © OpenStreetMap contributors 2025. Distributed under the Open Data Commons Open Database License (ODbL) v1.0.






#### 3.1 Zurich - Hardau

In Zurich, the REA and EC measurements were conducted on an antenna of 16.5 m height on top of a 95.3 m high-rise building, i.e., approximately 112 m above ground level at the site Zurich - Hardau (ICOS Station ID 'CH-Har', Table 2, Fig. C1). The building, called Hardau II, is located roughly 1.5 km northwest of the city center of Zurich, Switzerland (Fig. 1 a). It is surrounded by three similar buildings of lower height (66 m, 76 m, and 85 m). Apart from that, the average building height within a 1.5 km radius is  $13.3 \pm 8$  m. Located to the north are an industrial sector, railway lines, and busy arterial roads, to the west is a residential, green area with a cemetery, and to the southeast is an urban sector and the city center. The largest point source in the immediate vicinity, located 145 m southeast, is a district heating plant that uses natural gas.

During the first REA measurements in July 2022, different deadband settings ( $\delta = 0.3, 0.4, 0.7$ , and 0.8 with pre-set deadband) and averaging times (45, 60, 75 min) were tested (Table 2). With the pre-set deadband, in about 75 % of the REA measurements at least one of the reservoirs did not collect sufficient air to fill a flask. Therefore, a dynamic deadband with  $\delta = 0.7$  was implemented and has been used since the end of August 2022. This was better suited for variable wind conditions and increased the percentage of successful measurements to 75 %. Unfortunately, all samples collected between November 2022 and February 2023 had to be discarded due to a leak in the REA sampler, which was detected retrospectively. More details on the Zurich measurements are given in Kunz et al. (2025).

#### 3.2 Paris - Romainville

In Paris, the REA and EC systems were installed on an active telecommunications tower about 5 km northeast from the city center at the site Paris-Romainville (ICOS Station ID 'FR-Rmv', Table 2). The IRGASON and the gas inlets were mounted on a pylon, approximately 9 m above a wide ( $\sim 30$  m) platform (Fig. C1). The tower is located on a small hill in a densely urbanized area (Fig. 1 b).

Between July 2023 and April 2024, 66 of 384 successful and 498 scheduled REA measurements were analyzed in the laboratory. One sample was rejected due to abnormal  $^{12}$ C currents during  $^{14}$ C analysis at the accelerator mass spectrometer (AMS), as well as implausible measurement results, leaving 65 REA measurements with  $^{14}$ C and EC data (Table 2). Due to the massive structure of the tower and the resulting wind distortion effects, no samples were collected from wind directions between  $70^{\circ}$  and  $120^{\circ}$  N. For the vast majority of the analyzed samples, the mean wind direction was between  $180^{\circ}$  and  $225^{\circ}$  N. The deadband was initially scaled with  $\delta = 0.7$ , as in Zurich, but was increased to  $\delta = 0.9$  in October 2023 due to very small concentration differences between updrafts and downdrafts. With a pump speed of about  $7 \text{ l min}^{-1}$ , this was the maximum possible deadband width to collect sufficient air during a 60 min sampling period. Since the concentration differences were still close to the detection limit, the option for HREA was implemented in the logger program (Sect. 2.2) at the beginning of April 2024. To test the HREA method, nine samples were collected with H = 0.6. Due to technical problems with the MGA $^7$  in 2023, only EC measurements of the IRGASON are available for 2023. Between November 2023 and January 2024, the MGA $^7$  was dismantled for repairs and no REA measurements were conducted.

#### 3.3 Munich - Oberpostdirektion

From July 2024 to April 2025, REA measurements were carried out on a mast of an active telecommunications tower about 1.5 km northwest of the city center of Munich at the site Munich-Operpostdirektion (ICOS Station ID 'DE-Opd', Table 2). The tower has three platforms up to a height of 59 m and a mast on top, on which the IRGASON and the gas inlets were mounted at a height of 85 m (Fig. C1). In addition, two mid-cost sensor systems, which are based on the Non-Dispersive InfraRed CO<sub>2</sub> sensors GMP343, Vaisala Oyj, Vantaa, Finland, measured the CO<sub>2</sub> concentration at heights of 85 m and 48 m (part of the Munich mid-cost network ACROPOLIS (Aigner et al., 2025)). The tower is located in an area with many residential houses and other buildings (Fig. 1 c). To the southeast is the central railway station and behind it the historic city center. The largest point source, located approximately 200 m to the southeast, is a brewery.

Due to lack of space, the MGA<sup>7</sup> and the REA sampler were placed in the basement of the tower, requiring inlet lines of 100 m length. During the maintenance of the REA system prior to its installation in Munich, larger flushing pumps were installed (Sect. 2.2). The sampling flow rate was increased to approximately  $11 \text{ l min}^{-1}$ . With the increased flow rate, less time was needed to collect enough air for laboratory analysis, so a larger deadband ( $\delta = 1.1$ ) could be used. For summer afternoons with predominantly small CO<sub>2</sub> fluxes, a hyperbolic deadband with hole size H = 0.8 was used to increase the signal-to-noise ratio.

#### 4 Results and discussion


Reqular quality control tests of the REA system showed an overall good performance of the hardware (Appendix D). However, as the quality of the REA measurements varies depending on the micrometeorological conditions during sampling and the signal-to-noise ratios, the analyzed REA measurements were flagged as well-mixed measurements, low-turbulence and storage measurements, or were not considered further (Sect. 4.1). Section 4.2 presents an example, namely the results of six REA measurements conducted in Munich on 09 October 2024. To analyze and compare the results from all three cities, the ffCO<sub>2</sub> and nfCO<sub>2</sub> fluxes are compared with the net CO<sub>2</sub> fluxes, and the correlation between ffCO<sub>2</sub> and CO<sub>2</sub>, as well as spatial and temporal patterns, are investigated in Sect. 4.3. Furthermore, the representativeness of the results and the limitations of the methodology are examined. In Sect. 4.4, the potential of REA measurements with low turbulence or substantial storage fluxes is investigated. Finally, the absolute CO<sub>2</sub> and <sup>14</sup>CO<sub>2</sub>-based ffCO<sub>2</sub> concentrations of the REA flask samples are compared to marine background concentrations (Sect. 4.5).

## 4.1 Flagging of analyzed REA measurements

In Zurich, only 30 out of 87 REA measurements with  $^{14}$ C and EC data met the criteria finally considered, describing suitable well-mixed conditions (Table 3). Twelve samples were selected knowing that with  $u_* < 0.2 \text{ m s}^{-1}$  or  $|F_{\text{CO}_2, \text{strg}}| > 20 \text{ } \mu\text{mol m}^{-2} \,\text{s}^{-1}$  the measurements probably do not represent the surface fluxes during the sampling period. Most of these measurements with low-turbulence or storage flag were taken in the early morning and analyzed to obtain information on the composition of the nocturnal  $\text{CO}_2$  fluxes. As it was initially decided to relax the stationarity requirements due to the intermit-

**Table 3.** Number of REA measurements with ffCO<sub>2</sub> flux data and the percentages of measurements with ffCO<sub>2</sub> flux flagged as well-mixed, low-turbulence, storage or not considered further (Sect. 2.2.4). Quality control flag QC, beta coefficient  $\beta$ , and signal-to-noise ratio SNR are defined as in Table 1. Measurements not considered may be affected by multiple criteria.

| Number (percentage) of REA measurements | Zurich    | Paris     | Munich    |
|-----------------------------------------|-----------|-----------|-----------|
| with $ffCO_2$ flux data                 | 87        | 65        | 99        |
| with well-mixed flag                    | 30 (34 %) | 32 (49 %) | 78 (79 %) |
| with low-turbulence and/or storage flag | 12 (14 %) | 4 (6 %)   | 13 (13 %) |
| not considered                          | 45 (52 %) | 29 (45 %) | 8 (8 %)   |
| due to $QC = 2$                         | 22 (25 %) | 10 (15 %) | 3 (3 %)   |
| due to $\beta < 0.1$ or $\beta > 1$     | 13 (15 %) | 10 (15 %) | 1 (1 %)   |
| due to SNR $<100\%$                     | 28 (32 %) | 21 (32 %) | 5 (5 %)   |

tent nature of CO<sub>2</sub> fluxes in urban environments, 25 % of the periods did not meet the stationarity or well-developed turbulence 290 criteria. The  $\beta$  criterion was not considered in the selection of the flasks, but only 15 % of the measurements were affected. Excluding measurements with  $\beta < 0.1$ ,  $\beta > 1$  and QC = 2,  $\beta$  was  $0.44 \pm 0.14$  for a dynamically adjusted deadband width of  $0.7\sigma_w$ . This is slightly higher than the value of 0.39, which would be expected for a normally distributed timeseries with  $\delta = 0.7$  (Fotiadi et al., 2005b), but in good agreement with experimental data (e.g., Pattey et al., 1993; von der Heyden et al., 2022) (see Appendix B2). The main limitation of the Zurich REA measurements was a signal-to-noise ratio of < 100 %, 295 caused by the small  $\Delta^{14}$ C differences between updraft and downdraft samples compared to the mean measurement uncertainty of the Zurich samples of 1.8 % (\Delta notation according to Stuiver and Polach (1977)). In Paris, low-turbulence and storage measurements were mostly discarded. The  $\beta$  coefficient for  $\delta = 0.7$  was  $0.40 \pm 0.20$ , i.e., slightly smaller than in Zurich and in good agreement with theoretical expectations for normally distributed time series. Unfortunately, increasing  $\delta$  to 0.9 did not increase the concentration differences. For the selected measurements,  $\beta$  was even slightly larger on average (0.46  $\pm$  0.17, see 300 Appendix B2). As in Zurich, the main limitation of the measurements in Paris was a low signal-to-noise ratio. In Munich, the proportion of suitable measurements was significantly improved. The concentration differences were generally increased by a larger deadband width and HREA. The  $\beta$  coefficient was  $0.34 \pm 0.07$  for a deadband with  $\delta = 1.1$  and  $0.26 \pm 0.06$  in the case of HREA with H = 0.8, i.e., as expected much smaller than in Zurich and Paris (Appendix B2). At the same time, the  $\Delta^{14}\mathrm{C}$  measurement uncertainties were reduced by a new AMS from  $2.1\pm0.3$  % (Zurich samples with old AMS) to  $1.2\pm0.1$ 305 %, so that samples with SNR > 100 % could be selected. As in Zurich, low-turbulence and storage samples collected in the morning were deliberately selected to analyze the ffCO<sub>2</sub>/CO<sub>2</sub> ratio of nocturnal integrated fluxes. An overview of all REA measurements and their corresponding flags can be found at https://doi.org/10.5281/zenodo.17183699.

The results presented in Table 3 illustrate that the quality of a collected REA data set strongly depends on site-specific conditions such as flux strength or micrometeorological conditions, technical settings such as the deadband, and the data and







knowledge available during the campaign for the selection of suitable flask samples adapted to the scientific question. In our case, the largest number of high-quality  $ffCO_2$  flux data could therefore be collected in Munich.

#### 4.2 Example diurnal course

To illustrate the principle and to show an example of partitioning net fluxes of  $CO_2$  collected via EC into fossil and non-fossil  $CO_2$  flux components using REA, Fig. 2 presents data collected on 09 October 2024 in Munich. On that day, micrometeorological conditions were suitable for REA measurements and six flask pairs, sampled between 08:00 and 19:00 local time (UTC+2), were analyzed for  $^{14}CO_2$ . The hour-long sampling periods are highlighted in Fig. 2 in light blue.

The  $\mathrm{CO}_2$  concentration of ambient air, as measured by the two mid-cost sensor systems at 48 m above ground level (Fig. 2 a), follows the typical diurnal  $\mathrm{CO}_2$  cycle of a warm and sunny summer day (e.g., Stull, 1988; Lan et al., 2020). During night, the  $\mathrm{CO}_2$  concentration increases and a vertical concentration gradient with highest values close to the surface develops. As vertical mixing is suppressed ( $u_* \leq 0.2 \,\mathrm{m \ s^{-1}}$ , see Fig. 2 b), this can be attributed to surface emissions accumulating within the stable nocturnal boundary layer. After sunrise, friction velocity, temperature, and radiation increase (Fig. 2 b). As the radiative heating of the surface generates convective turbulent vertical motions, the vertical concentration gradient diminishes. The  $\mathrm{CO}_2$  concentration decreases at both heights first rapidly due to the entrainment of fresh air from higher altitudes, then more slowly as the depth of the atmospheric boundary layer stabilizes and changes in  $\mathrm{CO}_2$  concentration are primarily driven by the surface fluxes.

The continuous EC measurements (Fig. 2 d) show that the turbulent  $CO_2$  fluxes at 85 m height are approximately 10  $\mu$ mol m<sup>-2</sup> s<sup>-1</sup> in the early morning, increase after sunrise, and reach a maximum of more than 60  $\mu$ mol m<sup>-2</sup> s<sup>-1</sup> at noon, before they decrease again. This pattern is reflected in the net  $CO_2$  concentration differences between the sampled updraft and downdraft flasks, which are 3 ppm in the morning, 5.2 ppm at 12:00, and 2.2 ppm in the evening (Fig. 2 c). The <sup>14</sup>C-based  $\Delta$ ffCO<sub>2</sub> estimates (Fig. 2 c) indicate that during noon and in the evening, these net  $CO_2$  differences are entirely caused by fossil fuel emissions. Consequently, the ffCO<sub>2</sub> flux equals the net EC-based  $CO_2$  flux, while the nfCO<sub>2</sub> flux is approximately zero. In the morning and in the afternoon, on the other hand, the  $\Delta$ ffCO<sub>2</sub>/ $\Delta$ CO<sub>2</sub> ratio, and thus also the  $F_{\rm ffCO_2}/F_{\rm CO_2}$  ratio varies between 23 % and 43 %, indicating positive nfCO<sub>2</sub> fluxes of about 10 to 30  $\mu$ mol m<sup>-2</sup> s<sup>-1</sup>. Unfortunately, the  $\Delta$ ffCO<sub>2</sub> uncertainties for the REA measurements at 10:00 and 16:00 LT are unusually high due to technical issues during the <sup>14</sup>CO<sub>2</sub> AMS measurements in the subsequent lab analysis.

It must be noted that  $F_{\rm nfCO_2}\approx 0$  does not necessarily mean that there is no biospheric activity, but only that the positive fluxes (respiration + biofuels) approximately equal the photosynthetic uptake. Moreover, as discussed in Sect. 2.2.1, the EC and REA data represent the turbulent fluxes and are not corrected for changes in storage below the measurement height. This is particularly relevant for the measurement at 08:00, where  $u_* 

Figure 2. Visualization of EC and REA measurements on 09 October 2024 in Munich. Sampling periods of the six REA measurements are highlighted in blue. Arrows at the bottom of the plots indicate the mean horizontal wind direction and wind speed over 30 min. Day and night times are indicated by the gray bar. a)  $CO_2$  in situ measurements of the GMP343 at 85 m (= REA sampling height) and 48 m together with  $CO_2$  concentrations of the updraft and downdraft flask samples. b) 30 min averages of friction velocity  $u_*$ , photosynthetically active radiation PAR and air temperature  $T_{air}$  (†PAR was approximated by 1.7  $\mu$ mol J<sup>-1</sup> times the average incoming shortwave radiation). c)  $CO_2$  concentration differences between updraft and downdraft flask samples  $\Delta CO_2$  and their fossil and non-fossil components derived from the respective  $^{14}CO_2$  measurements. d) Continuous  $CO_2$  flux and  $CO_2$  storage flux estimates from EC measurements of the MGA<sup>7</sup> with 30 min averaging period. Blue bars indicate the mean net  $CO_2$  fluxes during the REA sampling periods, gray and green bars the respective fossil and non-fossil components derived from the flask concentration differences.

previous night. This highlights that the  $20 \ \mu mol \ m^{-2} \ s^{-1}$  threshold flags only the most extreme storage flux measurements, and that the flagging is not unambiguous, especially given the high uncertainty in the storage flux estimates in the morning. Since storage fluxes are usually largest in the morning, the well-mixed measurements are additionally analyzed for differences between measurements taken before and after 9:00 UTC (Sect. 4.3.3).

Overall, the measurements on 09 October 2024 in Munich indicate that the contribution of ffCO<sub>2</sub> emissions to the measured net CO<sub>2</sub> fluxes can vary significantly. However, <sup>14</sup>C-based ffCO<sub>2</sub> flux data are only available for a limited number of discrete time periods and often have uncertainties > 100 %.

#### 4.3 Partitioning of net CO<sub>2</sub> fluxes under well-mixed conditions

#### 4.3.1 Overview of sampling times and ffCO<sub>2</sub> vs. CO<sub>2</sub> fluxes from all three cities

As general overview over the REA measurements from Zurich, Paris, and Munich, Fig. 3 shows for each city the net  $CO_2$  fluxes during the selected REA sampling intervals with well-mixed conditions over the hour of the day and over the respective  $^{14}C$ -based  $ffCO_2$  flux estimates. In this way, the  $ffCO_2/CO_2$  flux ratios ( $R_{ffCO_2}$ ) as well as their temporal variability and representativeness can be classified and differences or similarities between the three cities and measurement campaigns can be analyzed qualitatively. A more quantitative analysis and dependencies on other parameters such as wind direction are discussed in Sect. 4.3.2 and Sect. 4.3.3.

In the right panels of Fig. 3, the 1:1 line marks the case when the net  $CO_2$  flux equals the ff $CO_2$  flux and the nf $CO_2$  flux is approximately zero. Accordingly, measurements above the 1:1 line have a net positive nf $CO_2$  flux component, while measurements below the line have a negative nf $CO_2$  component, i.e., photosynthesis has dominated. The magnitude of the nf $CO_2$  flux is indicated by the parallel dashed lines and the axes on the right, and the  $F_{nfCO_2}$  uncertainties by the vertical error bars. Due to error propagation of the ff $CO_2$  flux uncertainties, the nf $CO_2$  flux uncertainties are usually much larger than the uncertainties of the net  $CO_2$  fluxes, which are shown in the left panels. Negative ff $CO_2$  surface fluxes are unreasonable and are attributed to the limited resolution of small  $^{14}CO_2$  differences between updraft and downdraft samples (the error bars indicate the 1 $\sigma$  uncertainties). Nevertheless, the measurements are shown here, because they have a significant nf $CO_2$  component (SNR > 100 %).

For comparison, the median fluxes of the continuous EC  $CO_2$  measurements are shown in the left panels of Fig. 3 for both summer and winter. In this work, "summer" refers to the period from 15 July to 31 October, and "winter" to the period from 1 November to 15 April. This seasonal division of the measurement campaigns aligns roughly with the shift between European summer and winter time and with the change in local emissions due to heating degree days, and is consistent with other studies conducted in the same location during the same period (Hilland et al., 2025). Due to gaps in the MGA<sup>7</sup> data (Sect. 2.1), the IRGASON  $CO_2$  measurements are used, considering only high quality data with QC 

Figure 3. Overview of all REA measurements in Zurich, Paris, and Munich with well-mixed conditions. The color distinguishes summer measurements (15 July–31 October) and winter measurements (1 November–15 April). Left: Net  $CO_2$  flux  $F_{CO_2}$  during the REA sampling periods over the hour of the day. Error bars in x-direction indicate the length of the REA sampling period (mostly 60 min), error bars in y-direction the uncertainty of  $F_{CO_2}$  derived from the random uncertainty estimates of the EC measurements. The yellow and blue lines and shaded areas represent the medians and the interquartile ranges (IQR) of the continuous EC  $CO_2$  fluxes. Right:  $CO_2$  fluxes during the REA sampling periods compared to the  $^{14}C$ -based ff $CO_2$  fluxes. The areas with  $F_{\rm ffCO_2} 





median turbulent flux between 10:00 and 15:00 UTC is approximately zero. This means that in 50% of the considered time periods, negative  $CO_2$  fluxes, i.e., photosynthesis, were larger than positive  $CO_2$  fluxes through respiration and anthropogenic emissions. Large fluxes observed between 6:00 and 9:00 could be attributed to increased emissions during the morning rush hour and/or venting of nocturnally accumulated  $CO_2$ . In general, the median  $CO_2$  flux is highest in Munich, with largest differences compared to Zurich and Paris during summer daytime.

Compared to the median  $\mathrm{CO}_2$  fluxes, the fluxes during the selected REA sampling periods are often exceptionally high. This is caused by the systematic selection of flask pairs with large  $\mathrm{CO}_2$  concentration differences to increase the potential ff $\mathrm{CO}_2$  signal. In Zurich, all of the analyzed fluxes that exceeded the 75th percentile of the continuous EC fluxes (denoted as P0.75 in the following) were measured in winter and are almost entirely due to fossil fuel emissions. In Paris, there were only five REA measurements with  $F_{\mathrm{CO}_2} > \mathrm{P0.75}$ . As in Zurich, they were measured in winter, but they are not as clearly dominated by fossil fuel emissions as the large winter fluxes measured in Zurich. In Munich, turbulent fluxes > P0.75 were analyzed in both summer and winter, and most have a significant positive  $\mathrm{nfCO}_2$  component. Thus, while the large fluxes represent relatively rare conditions, the high signal-to-noise ratio (which was the main reason for analyzing them) allows observation of differences in the composition of the fluxes between the three cities (cf. Sect. 4.3.2 and Sect. 4.3.3).

REA measurements conducted in Zurich and Paris when  $CO_2$  fluxes were below P0.75 show positive and negative  $nfCO_2$  components of up to  $\pm 45~\mu mol~m^{-2}~s^{-1}$ . However, the uncertainties are large and there are very few summer measurements, as most of the measurements were flagged because of SNR < 100 %,  $\beta$  < 0.1 or  $\beta$  > 1. In Munich, on the contrary, the uncertainties are much smaller (see Table 4) and, except for a few measurements, all measurements show positive  $nfCO_2$  components. This means that respiration and biofuel emissions were generally larger than photosynthetic uptake. The latter is consistent with the observations from the continuous EC measurements that the net  $CO_2$  fluxes are highest in Munich and mostly positive throughout the year.

The correlation between the ffCO<sub>2</sub> and CO<sub>2</sub> fluxes is largest (0.68) for the Zurich winter measurements (Table 4). However, no clear correlation is observed when only the measurements with  $F_{\rm CO_2} 




Table 4. Mean uncertainties of the ffCO<sub>2</sub> fluxes  $F_{\rm ffCO_2}$  and the ffCO<sub>2</sub>/CO<sub>2</sub> flux ratios  $R_{\rm ffCO_2}$  of the REA measurements under well-mixed conditions in Zurich, Paris, and Munich. In addition, the Pearson correlation coefficients of the ffCO<sub>2</sub> and CO<sub>2</sub> fluxes and the mean air temperatures during the sampling periods are given. P0.75 denotes the 75th percentile of the continuous EC CO<sub>2</sub> fluxes.

| Variable                               | Unit                                                | Zurich          |                   | Paris           |                   | Munich             |                 |
|----------------------------------------|-----------------------------------------------------|-----------------|-------------------|-----------------|-------------------|--------------------|-----------------|
|                                        |                                                     | Summer          | Winter            | Summer          | Winter            | Summer             | Winter          |
| Number of REA measurements             | -                                                   | 6               | 24                | 8               | 24                | 40                 | 38              |
| Mean $F_{\mathrm{ffCO}_2}$ uncertainty | $\mu\mathrm{mol}\;\mathrm{m}^{-2}\;\mathrm{s}^{-1}$ | 8               | 12                | 9               | 13                | 6                  | 7               |
| Mean $R_{\mathrm{ffCO_2}}$ uncertainty | %                                                   | 126             | 59                | 86              | 71                | 31                 | 23              |
| Mean air temperature                   | °C                                                  | 18              | 9                 | 20              | 10                | 19                 | 6               |
| $Correlation(F_{ffCO_2}, F_{CO_2})$    | -                                                   | 0.47            | 0.68              | 0.43            | 0.22              | 0.54               | 0.34            |
| for $F_{\rm CO_2} < {\rm P0.75}$       | -                                                   | -0.19 $(N = 4)$ | 0.25 ( $N = 15$ ) | -0.86 $(N = 5)$ | 0.31 ( $N = 19$ ) | -0.02 ( $N = 26$ ) | 0.63 $(N = 27)$ |

#### 4.3.2 Spatial flux patterns and influence from point sources

This section examines the spatial distribution of the observed CO<sub>2</sub>, ffCO<sub>2</sub>, and nfCO<sub>2</sub> fluxes based on the mean horizontal wind directions during the well-mixed REA sampling periods. The aim is a first-order analysis of spatial patterns, which, if attributed to a specific land cover type, for example, is an important step towards generalizing the discrete flux measurements.

In Zurich, the net CO<sub>2</sub> fluxes observed with wind from the west are generally smaller than those with wind from the east

In Zurich, the net  $\mathrm{CO}_2$  fluxes observed with wind from the west are generally smaller than those with wind from the east (Fig. 4 a). The CO<sub>2</sub> fluxes > P0.75, which are clearly dominated by fossil fuel emissions (Fig. 4 b), are observed from about 70° N and 135° N. This is consistent with the high proportion of vegetated areas in west, in contrast to the city center, a district heating plant, and arterial roads in the east (Sect. 3.1). As emissions from point sources are generally not representative of the average fluxes in a city, and the comparison of measured and modeled point source emissions on an hourly basis is limited by uncertainties in the emissions inventory and transport models, we attempted to identify the REA measurements which were potentially influenced by emissions from the district heating plant. For this purpose, we considered the individual flux footprints according to Kljun et al. (2015) and the operating times of the district heating plant, which are known with a temporal resolution of 5 min. Based on these data, the measurements potentially influenced by emissions from the district heating plant were identified (represented in Fig. 4 in red, see Appendix E for details). Due to uncertainties and limitations in the footprint modeling, we also attempted to investigate the potential flux contribution from the district heating plant based on  $^{13}\mathrm{CO}_2$  observations.  $^{13}\mathrm{C}$  generally enables a distinction between  $\mathrm{CO}_2$  from natural gas, which is used in the district heating plant and has a  $^{13}$ C signature (in  $\delta$  notation) of about -40 % (Tans, 1981; Widory and Javoy, 2003), and that from liquid and solid fuel or biogenic fluxes with  $\delta^{13}C \approx -25 \%$  (Tans, 1981; Widory and Javoy, 2003; Bakwin et al., 1998). However, the Zurich REA flasks were not analyzed for <sup>13</sup>CO<sub>2</sub> by the high-precision ICOS Flask and Calibration Laboratory, but as a by-product of the <sup>14</sup>C extraction at the ICOS CRL, with an order of magnitude lower precision of about 0.2 %. Thus, although the potentially influenced samples showed influence from an isotopically lighter source, this was not significant within the

measurement uncertainties, and an unambiguous gas source attribution was not possible. A contribution from a gas source is therefore likely, but cannot be clearly attributed to individual measurements. Flagging of measurements with a potential contribution from emissions from the district heating plant was therefore based on footprint data alone.

Figure 4. Net  $CO_2$  fluxes (a), ff $CO_2$  fluxes (b), and nf $CO_2$  fluxes (c) with respect to the mean wind directions during the measurement intervals in Zurich with well-mixed conditions. The error bars represent the respective flux uncertainties. Measurements potentially influenced by emissions from a district heating plant to the southeast are indicated in red. P0.75 denotes the 75th percentile of the continuous EC  $CO_2$  fluxes at the respective hour of the day of the respective season. Indicated are also the directions of the arterial roads, the city center, and the district heating plant.

In Paris, measurements were primarily taken south-southwesterly wind. Due to the sparse data coverage, no spatial patterns can be investigated. There is no evidence of any distinct point-source emissions that could have affected the REA measurements. For completeness, the corresponding directional figures for Paris are shown in Appendix F.

Figure 5. Net  $CO_2$  fluxes (a), ff $CO_2$  fluxes (b), and nf $CO_2$  fluxes (c) with respect to the mean wind directions during the measurement intervals in Munich with well-mixed conditions. The error bars represent the respective flux uncertainties. Measurements potentially influenced by emissions from a district heating plant to the southeast are indicated in red. P0.75 denotes the 75th percentile of the continuous EC  $CO_2$  fluxes at the respective hour of the day. Indicated are also the directions of the brewery and the city center.





In Munich (Fig. 5), the highest CO<sub>2</sub> fluxes were measured when the wind came from southeast-east. Located in this direction are a brewery, the central railway station, and the historic city center ( $\sim 0.3, 1$ , and 2 km horizontal distance, respectively, see Sect. 3.3). Striking are the large  $nfCO_2$  fluxes of up to 50  $\mu$ mol m<sup>-2</sup> s<sup>-1</sup>. The fact that biospheric and human respiration fluxes are typically much smaller (e.g., Wu et al., 2022; Stagakis et al., 2023b, 2025) indicates a non-respiratory anthropogenic nfCO<sub>2</sub> source. Footprint analyses of the respective measurements, using the model of Kljun et al. (2015), show that the brewery was within the peak area of the flux footprint (Appendix E). Therefore, we assume that the large nfCO<sub>2</sub> emissions from the southeast-east result from a fermentation process (Elshani et al., 2018; Olajire, 2020). As there is no information available regarding operating times or the temporal emission profile of the brewery, all measurements with a substantial flux contribution from the brewery area, as estimated from the flux footprints, were considered to be potentially influenced by pointsource emissions (see Appendix E). Apart from measurements from the southeast, all  $nfCO_2$  fluxes  $> 20 \mu mol m^{-2} s^{-1}$  were measured in the early morning. As discussed in Sect. 4.2, this could indicate an unaccounted contribution from storage fluxes, further supported by the uncertainties in the distinction between low-turbulence, storage and well-mixed conditions (Sect. 4.4). The results thus emphasize that tall-tower measurements in urban environments can often be affected by individual point sources. In Zurich and Munich, most REA measurements with  $F_{\rm CO_2} > \rm P0.75$  are likely attributable to fossil emissions from a district heating plant and non-fossil emissions from a brewery, respectively. The measurements provide confidence in the EC and REA measurements as well as in the footprint analysis, and could be used to validate or refine bottom-up emission estimates of the respective point sources. Non-fossil CO<sub>2</sub> emissions from fermentation processes in breweries, for example, are usually not included in emission inventories. For the characterization of the usually smaller CO2 fluxes and the analysis of the biospheric nfCO<sub>2</sub> fluxes, however, these measurements need to be excluded.

#### 450 4.3.3 Mean ffCO<sub>2</sub>/CO<sub>2</sub> ratios and mean nfCO<sub>2</sub> fluxes

To generalize and quantify the results from the individual REA measurements, we analyze the mean ffCO<sub>2</sub>/CO<sub>2</sub> flux ratios  $\bar{R}_{\rm ffCO_2}$  and the mean magnitude of the nfCO<sub>2</sub> fluxes  $\bar{F}_{\rm nfCO_2}$  for each city. Due to the small number of measurements, it is not possible to fully account for the spatial and temporal variability. Based on the analyses in Sect. 4.3.1 and Sect. 4.3.2, we distinguish between summer and winter measurements, and exclude measurements which were potentially influenced by identified point-source emissions. Despite the flagging, it should be noted that due to the sampling method, the results may not be representative of the mean continuous CO<sub>2</sub> fluxes. This especially limits comparisons between the three cities.

If ffCO<sub>2</sub> and CO<sub>2</sub> fluxes were perfectly linearly correlated, the mean ffCO<sub>2</sub>/CO<sub>2</sub> ratios would be best described by the slope of an error-weighted total least squares regression line (Maier et al., 2024a). Due to the generally low correlations of the observed REA fluxes (Table 4), however,  $\bar{R}_{\rm ffCO_2}$  is determined as the error-weighted mean of the individual ffCO<sub>2</sub>/CO<sub>2</sub> ratios. To minimize the uncertainty, the individual  $R_{\rm ffCO_2}$  values are calculated directly from the flask measurements as  $\Delta \rm ffCO_2/\Delta CO_2$ , i.e., completely independent of the EC flux measurements (compare Eq. 3).  $R_{\rm ffCO_2} > 100$  % indicates a negative nfCO<sub>2</sub> flux, i.e., photosynthetic uptake, while  $R_{\rm ffCO_2} < 0$  % is physically unreasonable and only observed if  $\Delta \rm ffCO_2$  is slightly negative within its measurements uncertainties. In addition, the mean and variability of the nfCO<sub>2</sub> fluxes are examined. A z-test is used to evaluate whether the observations are significantly different from  $\bar{R}_{\rm ffCO_2} = 100$  % or  $F \rm nfCO_2 = 0$   $\mu \rm mol \, m^{-2} \, s^{-1}$  (signifi-



cance level of 0.05), i.e., completely fossil  $CO_2$  fluxes, taking into account the mean measurement uncertainties (Appendix G). To meet the assumption of normal distribution, only measurements with relative  $\Delta CO_2$  uncertainties « 1 are considered (most, but not all, of these samples are already excluded by the consideration of the signal-to-noise ratio as defined in Sect. 2.2.4).

Table 5. Error-weighted mean ffCO<sub>2</sub>/CO<sub>2</sub> flux ratio  $\bar{R}_{\rm ffCO_2}$  and error-weighted mean nfCO<sub>2</sub> flux  $F_{\rm nfCO_2}$  of the well-mixed REA measurements, excluding measurements in Zurich and Munich, which were potentially influenced by identified point-source emissions, and four measurements with  $\Delta {\rm CO}_2 < 0.4$  ppm. N is the number of samples. Stars indicate that, given the number of measurements and mean measurement uncertainties, the results are significantly different from  $\bar{R}_{\rm ffCO_2} = 100$  % or  $\bar{F}_{\rm nfCO_2} = 0$  µmol m<sup>-2</sup> s<sup>-1</sup>, respectively (\* p < 0.05, \*\*\* p < 0.01, \*\*\*\* p < 0.001).

|                     | $ar{R}_{	ext{ffCO}_2}$ [-] |                              |                              | $ar{F}_{ m nfCO_2}$ [ $\mu  m molm^{-2}s^{-1}$ ] |                             |                              |  |
|---------------------|----------------------------|------------------------------|------------------------------|--------------------------------------------------|-----------------------------|------------------------------|--|
|                     | Zurich                     | Paris                        | Munich                       | Zurich                                           | Paris                       | Munich                       |  |
| Summer measurements | $48 \pm 52 \%$ $(N = 3)$   | $-7 \pm 22 \%$ *** $(N = 8)$ | $47 \pm 4 \%$ *** $(N = 33)$ | $0 \pm 4$ $(N=3)$                                | $9.7 \pm 2.2$ *** $(N = 8)$ | $7.8 \pm 1.0$ *** $(N = 33)$ |  |
| Winter measurements | $92 \pm 11 \%$<br>(N = 16) | $80 \pm 10 \%$ * $(N = 23)$  | $76 \pm 4 \%$ *** $(N = 31)$ | $1.5 \pm 2.7$ $(N = 16)$                         | $2.7 \pm 2.1$ $(N = 23)$    | $5.3 \pm 1.1$ *** $(N = 31)$ |  |

In Zurich, no significant average  $nfCO_2$  signal (p-values > 0.05) was observed (Table 5). In summer, the mean  $nfCO_2/CO_2$  ratio was  $48\pm52$  % and the mean absolute  $nfCO_2$  flux was  $0\pm4$   $\mu$ mol  $m^{-2}$  s<sup>-1</sup>. The significance of the results was mainly limited by the small number of well-mixed measurements (N=3). In winter, the mean  $nfCO_2$  contribution of the Zurich samples was  $92\pm11$  %. To resolve the presumably small mean  $nfCO_2$  component, more measurements and/or smaller measurement uncertainties would have been necessary (see Appendix G).

In Paris, the eight selected summer samples showed mostly non-fossil  $CO_2$  contributions. The negative mean ff $CO_2$  ratio can be explained by the ff $CO_2$  flux uncertainties (compare Fig. 3), but a larger ff $CO_2$  contribution was expected. Note that most of the measurements were conducted in the early morning. Therefore, storage fluxes cannot be ruled out. However, due to the small number of samples, a further subdivision of the measurements into morning and afternoon measurements, for example, was not feasible. Similar to the Zurich measurements, the Paris measurements were generally more successful in winter than in summer due to larger signals. The mean ff $CO_2$  contribution in winter was  $80 \pm 10$  %, meaning that, on average, about 20 % of the observed  $CO_2$  emissions were due to positive nf $CO_2$  fluxes.

In Munich, the higher data quality and greater number of measurements enabled the detection of significant nfCO<sub>2</sub> contributions in both summer and winter. The larger nfCO<sub>2</sub> fluxes observed in summer compared to winter are primarily attributed to the measurements taken in the early morning during summer. When only 18 summer measurements taken after 9:00 UTC are considered,  $\bar{R}_{\rm ffCO_2}$  is  $64\pm6$ % and  $\bar{F}_{\rm nfCO_2}$  is  $5.6\pm1.3$  µmol m<sup>-2</sup> s<sup>-1</sup>, which is much smaller than for the early-morning measurements and comparable to the winter measurements. In winter, no significant differences were observed between measurements taken before and after 9:00 UTC. This could be explained by larger respiratory fluxes and nfCO<sub>2</sub> dominated storage




fluxes in the morning (Sect. 4.4) and larger photosynthetic uptake, i.e., negative  $nfCO_2$  fluxes in the afternoon. This temporal variability is larger in summer than in winter.

Overall, it is remarkable that the mean  $nfCO_2$  contributions are positive in all three cities, both in summer and in winter. Only a few measurements show a significant negative nfCO2 flux. This contrasts with various studies that estimated negative nfCO<sub>2</sub> fluxes in urban areas, particularly during the warm growing season but also during the cold dormant season (e.g., Wu et al., 2022; Miller et al., 2020). The positive nfCO<sub>2</sub> fluxes in our study could be explained, for example, by the low proportion of vegetated area within the flux footprints (Fig. 1). It should also be noted that the observed nfCO2 fluxes include human respiration. According to bottom-up estimates, the mean annual human respiration fluxes within a  $2 \times 2 \text{ km}^2$  square around the measurement sites are about 2.5  $\mu$ mol m<sup>-2</sup> s<sup>-1</sup>, or 10 % of the net CO<sub>2</sub> flux (Dröge et al., 2024). For comparison, the estimated human respiration flux in the footprint of the study by Wu et al. (2022) was only  $0.22 \, \mu \text{mol m}^{-2} \, \text{s}^{-1}$ . Human respiration could therefore account for a significant proportion of the observed nfCO2 fluxes. Moreover, due to the small number of analyzed samples and the systematic selection of samples with presumably large concentration differences, the results may be biased toward periods with positive nfCO<sub>2</sub> fluxes. As a further analysis, a 1:1 comparison of the REA fluxes with the emission inventories and biospheric models, taking into account the respective flux footprints, could be useful. As the example of the high nfCO2 fluxes from the direction of a brewery in Munich shows, the measurements could also be influenced by other anthropogenic nfCO<sub>2</sub> point sources. In Munich, for instance, there are also other, more distant breweries. Based on our flux footprint analysis, we excluded all measurements where one of these breweries could have impacted the measured flux. However, excluding these measurements had no significant impact on the results (not shown here). Consistent with the aforementioned studies, our measurements underscore the importance of nfCO<sub>2</sub> fluxes in urban areas.

#### 505 4.4 Low-turbulence and storage measurements

This section presents the ffCO<sub>2</sub>/CO<sub>2</sub> flux ratios of the low-turbulence and storage measurements conducted before 11:00 UTC. While turbulent fluxes measured when  $u_* < 0.2 \text{ m s}^{-1}$  and/or  $|F_{\text{CO}_2,\text{strg}}| > 20 \text{ }\mu\text{mol m}^{-2} \text{ s}^{-1}$  are unlikely to represent the instantaneous surface fluxes, their composition contains information about the relative strength of individual sources during the time of suppressed turbulence (Sect. 2.2.1). Therefore, measurements taken in the morning are assumed to contain information about the mean nocturnal emissions. As there is no photosynthetic activity at night, the nfCO<sub>2</sub> component represents the sum of soil respiration, plant respiration, human respiration, and biofuel burning. Since most low-turbulence and storage samples in Zurich were collected in winter and most low-turbulence and storage samples in Munich were collected in summer, the results are analyzed with respect to the mean night air temperature (Fig. 6).



Figure 6. ffCO $_2$ /CO $_2$  flux ratios ( $R_{\rm ffCO}_2$ ) of the low-turbulence and storage measurements taken before 11:00 UTC. The colors indicate whether  $|F_{\rm CO}_2, {\rm strg}| > 20~\mu{\rm mol~m^{-2}~s^{-1}}$  (storage flag) or  $|F_{\rm CO}_2, {\rm strg}| \le 20~\mu{\rm mol~m^{-2}~s^{-1}}$  (low-turbulence flag only). The error bars represent the measurement uncertainties. The x-axis shows the mean air temperature between 0:00 and 6:00 UTC on the respective days.

Although the samples were collected under very different conditions, i.e., in different cities, with variable contributions from surface and storage fluxes, at different times of the year, etc., the ffCO<sub>2</sub>/CO<sub>2</sub> flux ratios are mostly < 70 % and larger during cold temperatures than during warm temperatures. In Zurich, the error-weighted mean ffCO<sub>2</sub>/CO<sub>2</sub> ratio of the samples with night-time temperatures < 10 °C was  $68 \pm 7$  %. This indicates that the surface fluxes, as well as the accumulation of CO<sub>2</sub> in the stable nocturnal boundary layer, were primarily caused by fossil fuel emissions, e.g., due to building emissions, traffic, or industrial processes. However, there was also a substantial nfCO<sub>2</sub> contribution of about 30 % or more in winter. The samples collected in Munich with night temperatures > 10 °C show a mean ffCO<sub>2</sub>/CO<sub>2</sub> ratio of  $16 \pm 4$  %. The increased nfCO<sub>2</sub> contribution is most likely due to reduced traffic emissions at night, as well as no heating emissions and increased biospheric respiration in summer.

The results are in good agreement with other studies. Moriwaki et al. (2006) attributed the nocturnal build-up of  $CO_2$  in a suburban canopy layer in winter to the subsidence of (fossil) building emissions. Wu et al. (2022) observed nocturnal ff $CO_2/CO_2$  flux ratios in Indianapolis of  $\sim 66$  % in winter and  $\sim 33$  % in summer. In general, nocturnal net ecosystem exchange is found to be much larger in summer than in winter (e.g., Crawford and Christen, 2015; Stagakis et al., 2025).







#### 4.5 Comparison with regional CO<sub>2</sub> enhancements

While the REA flask measurements aimed to analyze turbulent ffCO $_2$  fluxes at the urban neighborhood scale, the absolute flask concentrations also contain information about the fossil and non-fossil CO $_2$  enhancements compared to clean background air and thus about the composition of CO $_2$  fluxes in a broader continental region, including other urban areas and regional emission sources. Following Levin et al. (2003), we calculate the ffCO $_2$  excess from the mean CO $_2$  and  $\Delta^{14}$ C values of the up- and downdraft REA sample pairs, using the corresponding concentration measurements at the European marine background station Mace Head on the western coast of Ireland as background concentrations, and assuming that the biogenic  $\Delta^{14}$ C signature equals the background concentration (see Appendix A2). Second-order effects, such as  $^{14}$ C-enriched heterotrophic respiration and nuclear contamination (Maier et al., 2023), were not considered because the necessary concentration footprints are only available until the end of 2023, and the corrections are negligible for our analysis. For details and an evaluation of these corrections on the Zurich measurements, we refer to Maier et al. (2023) and Appendix A2. The mean ffCO $_2$ /CO $_2$  ratios of the excess concentrations thus represent the average contributions of ffCO $_2$  emissions to the CO $_2$  fluxes on the trajectories between Mace Head and the three measurement sites. The results from all REA flask samples (the micrometeorological flagging criteria do not have to be applied for concentration measurements) are shown in Fig. 7. For clarity, the uncertainties of about 1 ppm are omitted, but are considered in the orthogonal regression.

While the concentration differences between updraft and downdraft samples, which were used to calculate the turbulent ffCO $_2$  fluxes (Eq. 3), are typically about 1 ppm, with a maximum of 14 ppm, the CO $_2$  and ffCO $_2$  enhancements compared to the background concentrations are significantly larger, especially in Zurich (median / maximum CO $_2$  enhancement of 14 / 123 ppm). Moreover, the regional CO $_2$  and ffCO $_2$  enhancements are much more correlated than the local turbulent fluxes and show a clear difference between summer and winter. For the summer samples, the mean ffCO $_2$ /CO $_2$  ratio obtained from orthogonal regression is 28 % for Zurich, 19 % for Paris, and 21 % for Munich, indicating that about 80 % of the net CO $_2$  enhancements in summer are due to non-fossil CO $_2$  emissions. For the winter samples, the average ratio is 63 % for Zurich, 51 % for Paris and 51 % for Munich, i.e., still much lower than the typical ffCO $_2$  flux contributions in the flux footprints (compare Sect. 4.3.3).

The results illustrate that the absolute  $CO_2$  concentrations at the measurement site are primarily driven by the background concentration (between 413 ppm and 435 ppm) and the regional  $CO_2$  fluxes integrated along the path from the marine background station to the urban area. In comparison to the local  $CO_2$  emissions, the regional fluxes are much more dominated by non-fossil  $CO_2$  emissions, in this case presumably biospheric respiration. The results agree well with those of Turnbull et al. (2015), who found that the  $ffCO_2$  enhancements measured in the city of Indianapolis with respect to a continental background station were two to three times higher than when a local background station directly upwind of the city was used. With a continental background, the  $ffCO_2$  enhancements accounted for only about 50 % of the net  $CO_2$  enhancement, whereas the local  $CO_2$  enhancement could be almost entirely explained by the  $ffCO_2$  contribution. Therefore, the  $CO_2$  fluxes analyzed in this paper represent only the local urban emissions and differ significantly from the net emissions in the surrounding area.

When analyzing  $CO_2$  concentrations, the choice of the background station is of great importance and must be adapted to the scientific question.

Figure 7.  $CO_2$  and  $ffCO_2$  excess concentrations ("xs") of the REA flask samples compared to concentration measurements at the European marine background station Mace Head. The pairs of updraft and downdraft measurements are connected by a line. For each site, the slope and the coefficient of determination  $R^2$  of a linear regression through the origin for the summer and winter measurements are given.







#### 5 Conclusions

A REA flask sampling system for <sup>14</sup>C-based estimation of ffCO<sub>2</sub> fluxes was operated alongside continuous EC CO<sub>2</sub> flux measurements on three urban tall towers in the cities of Zurich, Paris, and Munich for about nine months each. The analysis of 252 REA measurements was presented with regard to three research questions.

# 1. Potentials and limitations of <sup>14</sup>CO<sub>2</sub> REA measurements for CO<sub>2</sub> flux partitioning in cities

This study demonstrates the successful implementation of the REA method for  $^{14}\text{CO}_2$  measurements as a powerful technique for a purely observation-based separation of fossil and non-fossil  $\text{CO}_2$  fluxes. The Munich measurements show that with an improved technical setup and an adapted flask sampling and selection strategy, average  $\text{nfCO}_2$  fluxes of the order of 10% or  $3 \text{ } \mu \text{mol m}^{-2} \text{ s}^{-1}$  can be identified with a reasonable number of measurements (50 to 100). Assuming scalar similarity between  $\text{CO}_2$  and  $\text{^{14}CO}_2$ , the primary contributor to the overall flux partitioning uncertainty was the current  $\text{^{14}CO}_2$  measurement precision in the laboratory. At the given  $\text{CO}_2$  source strengths within the flux footprints of the chosen measurement sites, the signal-to-noise ratios were often below 100%. Situations with large fluxes are therefore favorable for the uncertainty-limited REA measurements and were preferentially selected for sample analysis. This systematic sample selection can introduce biases in the retrieved flux partitioning compared to the mean  $\text{CO}_2$  fluxes. Due to the complex, heterogeneous nature of urban environments, the micrometeorological requirements, and the costs and logistics associated with  $\text{^{14}CO}_2$  analyses, the  $\text{^{14}C}$ -based separation of ffCO<sub>2</sub> and nfCO<sub>2</sub> fluxes is limited to a small number of time periods and cannot be easily generalized.

#### 2. Indications for point sources and typical fossil and non-fossil CO2 flux compositions

In Zurich and Munich, sectorial high ffCO $_2$  or nfCO $_2$  fluxes indicated significant fossil and non-fossil anthropogenic CO $_2$  sources. Based on the respective flux footprints, these observations were potentially influenced by point-source emissions from a district heating plant in Zurich and a brewery in Munich, respectively. Excluding the measurements potentially influenced by point-source emissions, the mean ffCO $_2$ /CO $_2$  flux ratios of the analyzed winter measurements from the remaining urban emission mix were about 80 to 90 % at each of the three measurement sites, with average nfCO $_2$  fluxes of about 2  $\mu$ mol m $^{-2}$  s $^{-1}$  in Zurich and Paris and 5  $\mu$ mol m $^{-2}$  s $^{-1}$  in Munich. In Zurich and Paris, however, the average nfCO $_2$  components were within the uncertainties of the partitioning approach. In Munich, on the contrary, average nfCO $_2$  contributions were significantly larger than zero, especially in summer in the early morning and during conditions of low turbulence and/or changes in storage below the measurement height.

The conclusive results from the measurements potential influence by point-source emissions can be regarded as proof-of-concept for the  $\rm CO_2$  flux estimation based on EC measurements, flux partitioning based on  $\rm ^{14}CO_2$  REA measurements, and flux footprint modeling. At the same time, these measurements highlight the challenge of potential impact from point sources on tall-tower measurements in urban areas. Thus, to upscale tall-tower measurements to the city scale, compare them with bottom-up estimates, or to integrate them into inversion models, emission inventories and footprint models must represent point-source emissions (both fossil and non-fossil) and their temporal emission characteristics with high temporal and spatial resolution. The footprint models must also be capable of accounting for different emission heights and potential plume rise. The Munich

measurements indicate the importance of plant and soil respiration, human respiration, and non-respiratory anthropogenic 595 nfCO<sub>2</sub> emissions, especially during night. However, the representativeness of the selected REA measurements must be further analyzed. This again highlights the fundamental challenges of extrapolating local observations to derive emissions at the scale of an entire city.

#### 3. Compositions of local vs. regional CO2 fluxes

While the mean ffCO<sub>2</sub>/CO<sub>2</sub> flux ratios were about 80 % in winter and 50 % in summer, the CO<sub>2</sub> concentration enhancements 600 compared to marine background concentrations were in all three cities on average < 63 % in winter and < 28 % fossil in summer. This illustrates the locality of the urban flux footprint characterized by ffCO<sub>2</sub> emissions compared to the significantly larger continental concentration footprint, where biogenic fluxes dominate. A thorough selection of background stations is of great importance for the interpretation of urban CO<sub>2</sub> concentration enhancements.

#### 605 Outlook



In the future, the REA measurements could be used for a 1:1 comparison with hourly bottom-up estimates or as input (with uncertainties) to inversion models. As shown by Stagakis et al. (2023a), the assimilation of CO<sub>2</sub> flux observations from urban EC towers with very high spatiotemporal resolution information from urban bottom-up surface flux models has great potential to improve high-resolution bottom-up surface CO<sub>2</sub> flux model estimates. In future REA campaigns, a near real-time metric for identifying cases of atmospheric decoupling in cities under unstable conditions during the day could enable a more targeted selection or avoidance of samples influenced by storage fluxes based on the scientific question at hand. A multispecies analysis, including observations of co-emitted species such as CO, could allow for further attribution of emission sources and estimation of a continuous ffCO<sub>2</sub> flux record (e.g., Maier et al., 2024b; Hilland et al., 2025; Juchem et al., 2025).

Data availability. The raw data, the processed, quality-controlled fluxes, and the footprints used in this analysis are available from the ICOS Cities carbon portal https://citydata.icoscp.eu/portal/. Flags and comments on the individual REA measurements are provided at https: 615 //doi.org/10.5281/zenodo.17183699.

### Appendix A: ffCO<sub>2</sub> estimates

To estimate ffCO $_2$  concentrations, measured atmospheric  $\Delta^{14}C$  ( $\Delta$  notation according to Stuiver and Polach (1977)) and  $CO_2$ concentrations are considered as the sum of a background (bg), a fossil fuel (ff), a biofuel (bf), a nuclear (nuc), a stratospheric (strato), a respiratory (resp), a photosynthetic (photo), and an oceanic (oc) component (Turnbull et al., 2016; Maier et al., 2023):

$$c_{\text{meas}} = \sum c_i \tag{A1}$$

$$c_{\text{meas}} = \sum_{i} c_{i}$$

$$c_{\text{meas}} \Delta^{14} = \sum_{i} c_{i} \Delta^{14}_{i}.$$
(A2)


Here,  $\Delta^{14}$ C has been abbreviated by  $\Delta^{14}$  and i = bg, ff, bf, nuc, strato, resp, photo, oc. Although not all components from Eq. (A1) and Eq. (A2) are known, the budget equations allow, under certain assumptions, the calculation of ffCO<sub>2</sub> differences between updraft samples and downdraft samples from REA measurements as well as between individual measurements and a background concentration. This section shows the equations and values used in this study, while detailed derivations and justifications of the assumptions can be found in the relevant literature.

#### A1 Concentration differences between updraft and downdraft REA samples

Combining Eq. (A1) and Eq. (A2) and assuming that REA sample pairs differ only in their fossil fuel, non-fossil emissions (biofuel and respiration), and photosynthesis components, the difference in  $c_{\rm ff}$  between updraft and downdraft sample can be estimated via:

$$c_{\mathrm{ff}}^{\uparrow} - c_{\mathrm{ff}}^{\downarrow} = \frac{1}{\Delta_{\mathrm{photo}}^{14} - \Delta_{\mathrm{ff}}^{14}} \left[ c_{\mathrm{meas}}^{\uparrow} (\Delta_{\mathrm{photo}}^{14} - \Delta_{\mathrm{meas}}^{14}^{\uparrow}) - c_{\mathrm{meas}}^{\downarrow} (\Delta_{\mathrm{photo}}^{14} - \Delta_{\mathrm{meas}}^{14}^{\downarrow}) + (c_{\mathrm{nf}}^{\uparrow} - c_{\mathrm{nf}}^{\downarrow}) (\Delta_{\mathrm{nf}}^{14} - \Delta_{\mathrm{photo}}^{14}) \right]. \tag{A3}$$

We follow Maier et al. (2023) to account for the second-order effects of non-fossil  $^{14}\mathrm{CO}_2$  fluxes and assume that a) the  $^{14}\mathrm{CO}_2$  signature of photosynthetic fluxes equals the mean of the updraft and downdraft flasks, b) respiration fluxes are enriched by  $25\pm12$  ‰ compared to the mean atmospheric signature in the respective summer (July–September), and c) that the  $\mathrm{CO}_2$  concentration difference between updraft and downdraft flasks resulting from respiration and biofuels can be roughly accounted for with  $5\pm5$  ppm as an upper limit. Table A1 shows the assumptions and values for  $\Delta_{\mathrm{photo}}^{14}$ ,  $\Delta_{\mathrm{nf}}^{14}$ , and  $\Delta c_{\mathrm{nf}}$  used for the Zurich, Paris, and Munich measurements. Details and an analysis of the corresponding uncertainties can be found in Kunz et al. (2025).

Table A1. Variables used to estimate  $c_{\rm ff}^{\uparrow} - c_{\rm ff}^{\downarrow} = \Delta {\rm ffCO_2}$ .  $\Delta_i^{14}$  denote the  $\Delta^{14}{\rm C}$  values of fossil fuels (ff), photosynthetic (photo) and nonfossil emissions (nf)  ${\rm CO_2}$ , and flask measurements (meas).  $\overline{\Delta_{\rm meas}^{14}} = 0.5 \cdot (\Delta_{\rm meas}^{14}{}^{\uparrow} + \Delta_{\rm meas}^{14}{}^{\downarrow})$  denotes the mean of the updraft and downdraft samples, which is different for each REA sampling. The atmospheric signature during  ${\rm CO_2}$  uptake of the biosphere  $\overline{\Delta_{\rm atmo}}$  is estimated by the mean  $\Delta_{\rm meas}^{14}$  value in summer (July to September 2022/2023/2024 in the case of the Zurich/Paris/Munich campaign). Also given are the specific values derived for the measurement campaigns in each city.

| Variable                                        | Unit                                                | Approximation                                 | Zurich value                                  | Paris value                               | Munich value                              |
|-------------------------------------------------|-----------------------------------------------------|-----------------------------------------------|-----------------------------------------------|-------------------------------------------|-------------------------------------------|
| $\Delta_{ m ff}^{14}$                           | %0                                                  | -1000                                         | -1000                                         | -1000                                     | -1000                                     |
| $\Delta_{ m photo}^{14}$                        | $% c = \frac{1}{2} \left( \frac{1}{2} \right)^{-1}$ | $\overline{\Delta_{ m meas}^{14}}$            | $\overline{\Delta_{\mathrm{meas}}^{14}}\pm10$ | $\overline{\Delta_{\rm meas}^{14}}\pm 10$ | $\overline{\Delta_{\rm meas}^{14}}\pm 10$ |
| $\Delta_{ m nf}^{14}$                           | $% c = \frac{1}{2} \left( \frac{1}{2} \right)^{-1}$ | $\overline{\Delta_{\mathrm{atmo}}^{14}}$ + 25 | $9 \pm 16$                                    | $14\pm14$                                 | $5 \pm 14$                                |
| $c_{ m nf}^{\uparrow} - c_{ m nf}^{\downarrow}$ | ppm                                                 | $\sim \overline{\Delta \mathrm{CO}_2}$        | $5\pm5$                                       | $5\pm5$                                   | $5\pm5$                                   |

#### A2 Concentration differences between REA flasks and a marine background station

Approximating  $\Delta^{14}_{\rm photo}$  by  $\Delta^{14}_{\rm bg}$ , the ffCO<sub>2</sub> concentration compared to clean background air can be calculated from Eq. (A1) and Eq. (A2) according to Maier et al. (2023):



$$c_{\rm ff} = c_{\rm meas} \cdot \frac{\Delta_{\rm bg}^{14} - \Delta_{\rm meas}^{14}}{\Delta_{\rm bg}^{14} - \Delta_{\rm ff}^{14}} + c_{\rm meas} \cdot \frac{\Delta_{\rm nuc}^{14}}{\Delta_{\rm bg}^{14} - \Delta_{\rm ff}^{14}} + c_{\rm resp} \cdot \frac{\Delta_{\rm resp}^{14} - \Delta_{\rm bg}^{14}}{\Delta_{\rm bg}^{14} - \Delta_{\rm ff}^{14}}$$
(A4)

As described in detail in Maier et al. (2023), the background concentrations can be estimated from measurements at the ICOS station Mace Head (MHD) on the western coast of Ireland. Nuclear contributions can be modeled using a dedicated Jupyter notebook from the ICOS Carbon Portal (https://www.icos-cp.eu/data-services/tools/jupyter-notebook, last access 20 September 2025). Respiratory concentrations can be obtained using the Vegetation Photosynthesis and Respiration Model (VPRM, Mahadevan et al. (2008)) in combination with the Stochastic Time-Inverted Lagrangian Transport model (STILT, Lin et al. (2003)). However, STILT simulations require meteorological input fields, which are to date only available until the end of 2023. Therefore, the nuclear and respiratory corrections (last two terms in Eq. A4) were neglected in our analysis (Levin et al., 2003). Figure A1 compares the Zurich results with and without the corrections. The slopes of the linear regressions differ about 4 %. Part of this difference is due to the exclusion of one summer and one winter sample that could have been affected by a revision of a nuclear facility. For a qualitative comparison of local ffCO<sub>2</sub> REA fluxes and regional ffCO<sub>2</sub> concentration enhancements, however, the nuclear and respiratory corrections are considered negligible.

Figure A1. Comparison of concentration enhancements of the Zurich REA samples with respect to MHD (a) without corrections and (b) with corrections for nuclear contamination and  $^{14}$ C-enriched respiration in the ffCO<sub>2</sub> estimation.  $R^2$  is the coefficient of determination of the orthogonal regression, N the number of samples considered.



#### Appendix B: Flagging criteria for analyzed REA measurements

#### B1 Stationarity and well-developed turbulence

As with any turbulent trace gas flux measurement method, stationarity and and well-developed turbulence are prerequisites for taking REA measurements (Rinne et al., 2021). We use the 0-1-2 quality control flagging scheme according to Mauder and Foken (2004), which labels "0" as high quality fluxes, "1" as medium quality fluxes, and "2" as poor quality fluxes, based on the steady state test and the developed turbulence test (Foken and Wichura, 1996). For the usually 60 min long REA measurements, the maximum of the 30 min EC averaging periods is considered.

#### B2 $\beta$ coefficients

Figure B1 shows the  $CO_2$  flux  $F_{CO_2}$  with respect to the product of the air density  $\rho$ , the standard deviation of the vertical wind velocity  $\sigma_w$ , and the  $CO_2$  difference between updraft and downdraft flasks of all REA flask samples collected in Zurich, Paris, and Munich. The high correlation between the EC-based  $F_{CO_2}$  and the REA-based  $\sigma_w \overline{\rho_m} \Delta CO_2$  shows the high quality of both measurement methods. According to Eq. (2), the slope of a linear fit corresponds to the  $\beta$  coefficient. If the vertical wind velocity w were normally distributed and the regression on the  $CO_2$  concentration were linear,  $\beta$  would depend only on the deadband width  $\delta$ . Then all data points with the same  $\delta$  would fall on a line with a slope of  $\beta=0.627$  for  $\delta=0$  and smaller slopes for larger  $\delta$  (Grönholm et al., 2008). Deviations from this line indicate deviations from a Gaussian distribution. Since differences between individual measurements were found to be larger than differences between different scalars (Grönholm et al., 2008; Pattey et al., 1993), this is taken into account by calculating  $\beta$  for each sampling period individually according to Eq. (2). However, Eq. (2) is unstable for  $\Delta CO_2$  close to zero, and  $\beta < 0.1$  or  $\beta > 1$  indicate non-ideal sampling conditions for REA measurements, e.g., due to skewness and kurtosis of the w time series or a linear drift leading to an unequal distribution of sampling times into the updraft and the downdraft reservoirs (Fotiadi et al., 2005a; Grönholm et al., 2008). Following Hensen et al. (2009) and Osterwalder et al. (2016), we only analyze measurements with  $0.1 \le \beta \le 1$ .

Figure B1.  ${\rm CO_2}$  flux  $F_{{\rm CO_2}}$  vs. the standard deviation of the vertical wind velocity  $\sigma_w$  times the mean molar air density  $\overline{\rho_m}$  and the  ${\rm CO_2}$  concentration difference between updraft and downdraft flasks of all REA flask samples collected in Zurich, Paris, and Munich. The colored dashed lines correspond to a linear regression of the well-mixed measurements (including measurements with SNR < 100 %, only if N > 5) with slope  $\beta_{\rm fit}$ .

Table B1.  $\beta$  coefficients determined from the well-mixed measurements (including measurements with SNR < 100 %) from a linear regression of  $F_{\text{CO}_2}$  and  $\Delta$ CO<sub>2</sub> ("Fit", see Fig. B1) compared to the mean and standard deviation of the individually calculated  $\beta$  values (Eq. 2). In addition, the expected values for a normally distributed w and CO<sub>2</sub> timeseries are given (Fotiadi et al. (2005b), no value for H = 0.8 found in the literature). N denotes the number of samples considered.

| Deadband width    | City   | N  | Mean $\pm$ std  | Fit             | Gauss |
|-------------------|--------|----|-----------------|-----------------|-------|
| Linear $(\delta)$ |        |    |                 |                 |       |
| 0.7               | Zurich | 62 | $0.44 \pm 0.14$ | $0.39 \pm 0.01$ | 0.39  |
| 0.7               | Paris  | 20 | $0.40 \pm 0.20$ | $0.38 \pm 0.04$ | 0.39  |
| 0.9               | Paris  | 36 | $0.46 \pm 0.17$ | $0.40 \pm 0.02$ | 0.34  |
| 1.1               | Munich | 88 | $0.34 \pm 0.07$ | $0.33 \pm 0.01$ | 0.30  |
| Hyperbolic $(H)$  |        |    |                 |                 |       |
| 0.8               | Munich | 8  | $0.26 \pm 0.06$ | $0.24 \pm 0.02$ | ?     |

#### 675 B3 Signal-to-noise ratio


The calculation of fluxes based on REA measurements from Eq. (1) requires that the concentration difference between updraft and downdraft samples is greater than the measurement uncertainty. Otherwise, it is unclear whether the flux was actually small or whether it was a measurement error (Fotiadi et al., 2005a). In our case of separating net  $CO_2$  fluxes into fossil and non-fossil components, we consider the relative uncertainties of both ff $CO_2$  and nf $CO_2$  fluxes and discard samples only if both are > 100 %, otherwise the results would be biased toward large ff $CO_2$  fluxes. For this purpose, we define the signal-to-noise ratio (SNR) as the minimum of the relative uncertainties of the ff $CO_2$  and the nf $CO_2$  fluxes. Examples are shown in Fig. B2.

Figure B2. Three examples of REA measurements with signal-to-noise ratio SNR > 100 % (a) and SNR < 100 % (b). SNR is defined as the minimum of the relative uncertainties of the ffCO<sub>2</sub> and the nfCO<sub>2</sub> fluxes.







#### **B4** Friction velocity and storage fluxes

During or after time periods of low turbulence, the measurement system may be decoupled from the surface so that the eddy flux is no longer representative of the local surface flux (Aubinet et al., 2012a). Instead, the measured flux will also contain non-turbulent flux components. These components can be caused by changes in storage below the measurement height or by turbulence generated at elevated layers by high wind shear, for example (low-level jets (e.g., Prabha et al., 2007)). The composition of these non-turbulent fluxes will be largely determined by the surface fluxes prior to the measurement period. For example, the  $ffCO_2/CO_2$  ratio of a storage flux during the break up of the nocturnal boundary layer in the morning will approximately reflect the  $ffCO_2/CO_2$  ratio of the integrated nocturnal  $CO_2$  emissions. Due to reduced anthropogenic emissions at night, this nocturnal ratio is assumed to be lower than the  $ffCO_2/CO_2$  ratio of the surface fluxes during the measurement period (e.g., morning rush hour). Consequently, the mean  $ffCO_2/CO_2$  ratio of the integrated nocturnal  $CO_2$  emissions is assumed to be smaller than the measured  $ffCO_2/CO_2$  ratio. To identify the measurement periods of low turbulence and/or changes in the storage below the measurement height, we consider two quantities: the friction velocity  $u_*$  and the storage flux estimated from the EC measurements.

As the assumption of well-developed turbulence for EC is often not fulfilled during periods of low friction velocity,  $u_*$  is commonly used as criterion to filter EC fluxes (Aubinet et al., 2012a). Although friction velocities tend to be greater in cities due to enhanced mechanical forcing, for example, the use of a  $u_*$  filter has also proved useful in many urban studies (e.g., Wu et al., 2022; Vogel et al., 2024; Hilland et al., 2025). As the continuous EC  $CO_2$  fluxes showed a systematic decrease in flux magnitude at  $u_* < 0.2 \text{ m s}^{-1}$  (Hilland et al., 2025), these periods were flagged. Since  $u_*$  also becomes small during strong convective events, the  $u_*$  criterion was only applied to periods with stability parameter  $\zeta > -15.5$  (this threshold was chosen based on the stability range where the footprint model by Kljun et al. (2015) is applicable).

Although storage fluxes, i.e., changes in the mean  $CO_2$  concentrations within the air volume below the measurement height, are mostly negligible at higher  $u_*$ , a storage flux correction is usually applied and recommended for EC flux measurements (Crawford and Christen, 2014). Since we cannot apply such a storage correction to the REA ff $CO_2$  fluxes (see Sect. 2.2), measurements with large storage fluxes were flagged. The threshold was set to  $|F_{CO_2, strg}| > 20 \, \mu mol \, m^{-2} \, s^{-1}$ , which is relatively large compared to the median value of the continuous EC measurements of about 3  $\mu mol \, m^{-2} \, s^{-1}$ . Due to the limited number of analyzed REA samples and the large uncertainties in storage flux estimation, only the most extreme measurements were flagged. The resulting uncertainties are discussed in the text, and the results are analyzed with respect to differences between measurements taken before and after 9:00 UTC (for Munich only, due to the small number of measurements in Zurich and Paris, see Sect. 4.3.3).

Appendix C: Tall-tower installations

# a) Zurich

# b) Paris

# c) Munich

**Figure C1.** Photos of the measurement sites in Zurich, Paris, and Munich. The black arrows indicate the height at which the IRGASON and the gas inlets (two REA inlets with fast-response valves for updrafts and downdrafts, one inlet for REA quality control tests, one inlet for MGA<sup>7</sup> measurements) were mounted. Pictures from Pekka Pelkonen (ICOS RI), Pedro Henrique Herig Coimbra (INRAE), and Reiter Antennenbau-Energietechnik GmbH.

## Appendix D: Quality control of the REA system

In all three cities, quality control tests conducted approximately once a month showed an overall good agreement between the  $CO_2$  concentration of flasks sampled through the updraft and, in parallel, through the downdraft lines without switching of the valves (Table D1). The  $CO_2$  difference between these quality control flask pairs and air samples collected simultaneously through a third intake line directly into the flask sampler was slightly higher ( $\pm 0.1$  ppm on average). This difference can be partly attributed to the fact that with direct sampling, the weighting of the  $CO_2$  concentration over the sampling period is not completely homogeneous, leading to larger deviations if the  $CO_2$  concentration has a large variability or a trend (Levin et al., 2020; Kunz et al., 2025). It can therefore be assumed that biases between updraft and downdraft sampling are negligible.

For the analyzed REA sample pairs, the measured CO<sub>2</sub> differences between updraft and downdraft samples agree well with the CO<sub>2</sub> difference estimates from both concurrent and continuous open-path IRGASON and closed-path MGA<sup>7</sup> measurements (Table D1). As discussed in Kunz et al. (2025), a 0.2 ± 0.3 ppm difference between flask and IRGASON measurements in Zurich could be partly attributed to the fact that the IRGASON CO<sub>2</sub> dry molar fractions were derived from a CO<sub>2</sub> density output that does not properly account for high-frequency fluctuations in air temperature in the sensing path, because the ambient temperature measured by an EC100 slow-response temperature probe was used in the conversion of the absorption measurements to CO<sub>2</sub> density. Since 13 April 2024 (end of Paris measurements), an updated logger program records the CO<sub>2</sub> measurements using a fast-response temperature of the ultrasonic anemometer. The slightly smaller ΔCO<sub>2</sub> estimates from the MGA<sup>7</sup> in Munich may be due to high-frequency attenuation caused by the long intake lines affecting the MGA<sup>7</sup> (100 m vs. approximately 30 m in Zurich and Paris). Nevertheless, the overall good agreement between flask and in situ measurements indicates that the system was operating as intended and that uncertainties due to the sampling process are negligible. As shown in Kunz et al. (2025) for the Zurich measurements, the ffCO<sub>2</sub> flux uncertainties are dominated by the <sup>14</sup>C measurement precision.

**Table D1.** Means and standard deviations of the  $CO_2$  differences between quality control flasks sampled without switching of the valves (all-valves-open tests) through the updraft  $(CO_{2,qc^{\dagger}})$ , the downdraft  $(CO_{2,qc^{\dagger}})$  and a direct line  $(CO_{2,qc^{\dagger}})$ . Furthermore, the  $CO_2$  concentration  $\Delta CO_2$  between updraft and downdraft flasks collected during the actual REA measurements are compared to estimates from the 20 Hz in situ measurements of the IRGASON and the MGA<sup>7</sup>. For the latter, only IRGASON measurements with  $CO_2$  signal strength > 90 % and only MGA<sup>7</sup> measurements with good spectral fit of the  $CO_2$  laser are considered.

|                                                                                            | Zurich                         | Paris                        | Munich                        |  |
|--------------------------------------------------------------------------------------------|--------------------------------|------------------------------|-------------------------------|--|
| All-valves-open tests                                                                      |                                |                              |                               |  |
| $\mathrm{CO}_{2,\mathrm{qc}^\uparrow} - \mathrm{CO}_{2,\mathrm{qc}^\downarrow}$ [ppm]      | $-0.007 \pm 0.023$ $(N = 6)$   | $0.016 \pm 0.026$<br>(N = 7) | $-0.016 \pm 0.044$ $(N = 11)$ |  |
| $CO_{2,\overline{qc}} - CO_{2,qc \ direct}$ [ppm]                                          | $0.12 \pm 0.14$ $(N = 6)$      | $0.13 \pm 0.37$ $(N = 7)$    | $-0.14 \pm 0.18$ ( $N = 11$ ) |  |
| Flask - in situ comparison                                                                 |                                |                              |                               |  |
| $\Delta \mathrm{CO}_{2,\mathrm{flasks}}$ - $\Delta \mathrm{CO}_{2,\mathrm{IRGASON}}$ [ppm] | $0.21 \pm 0.3$<br>( $N = 85$ ) | $0.07 \pm 0.44$ $(N = 55)$   | $0.23 \pm 0.36$<br>(N = 92)   |  |
| $\Delta \mathrm{CO}_{2,\mathrm{flasks}}$ - $\Delta \mathrm{CO}_{2,\mathrm{MGA}^7}$ [ppm]   | $0.01 \pm 0.20$ $(N = 64)$     | $-0.03 \pm 0.26$<br>(N = 31) | $0.07 \pm 0.24$ $(N = 86)$    |  |

#### Appendix E: Point-source emissions in Zurich and Munich

In Zurich, emissions from a district heating plant (natural gas) are likely to have influenced the REA measurements when the district heating plant was operating and within the peak area of the flux footprint (Sect. 3.1 and Sect. 4.3.2). To identify the potentially affected REA measurements, the flux contributions from a  $40 \times 40$  m<sup>2</sup> area centered around the chimney of the district heating plant were estimated based on the footprint model by Kljun et al. (2015) (Fig. E1 a, left). For this purpose, two 30 min footprints were averaged for each REA measurement. There were three measurements in which the modeled footprint was in the direction of the district heating plant, but the contribution from the considered area was zero due to the finite distance of the peak contribution from the measurement site and the immediate proximity of the district heating plant to the tower ( $\sim 150$  m) (Fig. E1 a, right). Since the footprint model does not account for the height of the emissions (chimney of  $\sim 30$  m) and since  $CO_2$  spikes observed in the continuous concentration measurements indicate an influence from the point source, we assume that these three measurements could nevertheless have been influenced by the district heating plant. The operating times of the three burners of the district heating plant are known with a temporal resolution of 5 min.

Analogously, all Munich REA measurements in which the flux footprint contribution from the area where the brewery is located was > 3.5 %, were considered to be potentially influenced by emissions from the brewery (Fig. E1 b). In Munich, neither the operating times nor the exact location of the emission source is known.

**Figure E1.** Relative flux contributions from the areas where the district heating plant (Zurich) and the brewery (Munich) are located based on the flux footprints of the well-mixed REA measurements. Map data from © OpenStreetMap contributors 2025. Distributed under the Open Data Commons Open Database License (ODbL) v1.0.

# **Appendix F: Spatial flux patterns in Paris**

Figure F1. Net  $CO_2$  fluxes (a), ff $CO_2$  fluxes (b), and nf $CO_2$  fluxes (c) with respect to the mean wind directions during the measurement intervals in Paris with well-mixed conditions. The error bars represent the respective flux uncertainties. P0.75 denotes the 75th percentile of the continuous EC  $CO_2$  fluxes at the respective hour of the day. Indicated is also the direction of the city center.

750

755

### **Appendix G: Z-tests**

Since the relatively large positive and negative  $nfCO_2$  fluxes observed in Zurich and Paris for fluxes  $< 30~\mu mol~m^{-2}~s^{-1}$  could not be sufficiently explained by temperature, radiation or other variables, it was investigated to what extent the results could be caused by measurement uncertainties alone and whether the available data sets show a significant difference to the naive assumption of purely fossil fluxes in the city. For this purpose, a z-test was used to calculate the probability of measuring the observed error-weighted mean  $ffCO_2/CO_2$  ratios and mean  $nfCO_2$  flux under the null hypotheses of entirely fossil fluxes, i.e.,  $\bar{R}_{ffCO_2} = 100~\%$  or  $\bar{F}_{nfCO_2} = 0~\mu mol~m^{-2}~s^{-1}$ , given the mean measurement uncertainties (Table 4). The null hypothesis was rejected if the p-value was less than the significance level of 0.05. Since the z-test assumes a normal distribution of the observed variables, measurements with  $\Delta CO_2$  less than the measurement uncertainty of about 0.04 ppm were excluded to avoid extreme values in the  $ffCO_2/CO_2$  ratio. In addition, we determined the minimum effect, i.e., the minimum deviation from the null hypothesis that would be required to correctly reject the null hypothesis at a 0.05 significance level and 80 % power. Smaller deviations from  $\bar{R}_{ffCO_2} = 100~\%$  or  $\bar{F}_{nfCO_2} = 0~\mu mol~m^{-2}~s^{-1}$  could not be detected with the given number of samples and measurement uncertainties. The number of samples required to detect an assumed difference in the mean  $ffCO_2/CO_2$  ratio of 10 % or an assumed mean  $nfCO_2$  flux of 3  $\mu mol~m^{-2}~s^{-1}$  was also determined. Note that a constant  $ffCO_2/CO_2$  ratio is not compatible with a constant  $nfCO_2$  flux. However, both are possible conceptual models that are analyzed here. Table G1 and Table G2 show the results for the well-mixed measurements, divided into summer and winter measurements.

Table G1. Analysis of the ffCO<sub>2</sub>/CO<sub>2</sub> ratios of the well-mixed measurements, excluded those likely influenced by point source emissions and four measurements with  $\Delta {\rm CO}_2 < 0.4$  ppm. N denotes the number of measurements,  $\bar{R}_{\rm ffCO_2}$  the error-weighted mean ffCO<sub>2</sub>/CO<sub>2</sub> ratio, and  $\bar{\delta R}_{\rm ffCO_2}$  the mean measurement uncertainty of the ratios. The p-values describe the probabilities of observing the measured mean ratio under the assumption (null hypothesis) that  $\bar{R}_{\rm ffCO_2} = 100$  % and that deviations are solely due to measurement uncertainty. In addition, the minimum deviation from  $\bar{R}_{\rm ffCO_2} = 100$  % required to reject the null hypothesis at a significance level of 0.05 (minimum effect) and the number of samples required to detect a deviation from the null hypothesis of 10 % at significance level of 0.05 and a power of 80 % ( $N_{10}$ ) is given.

| Variable                                    | Zurich      |           | Paris     |           | Munich   |          |
|---------------------------------------------|-------------|-----------|-----------|-----------|----------|----------|
|                                             | Summer      | Winter    | Summer    | Winter    | Summer   | Winter   |
| $\overline{N}$                              | 3           | 16        | 8         | 23        | 33       | 31       |
| $ar{R}_{\mathrm{ffCO_2}}$ [%]               | $48 \pm 52$ | $92\pm11$ | $-7\pm22$ | $80\pm10$ | $47\pm4$ | $76\pm4$ |
| $\overline{\delta R}_{\mathrm{ffCO}_2}$ [%] | 106         | 70        | 86        | 63        | 33       | 25       |
| p-value                                     | 0.3         | 0.4       | 

Table G2. Analysis of the nfCO<sub>2</sub> fluxes of the well-mixed measurements, , excluded those likely influenced by point source emissions and four measurements with  $\Delta {\rm CO}_2 < 0.4$  ppm. N denotes the number of measurements,  $\bar{F}_{\rm nfCO_2}$  the error-weighted mean nfCO<sub>2</sub> flux, and  $\overline{\delta F}_{\rm nfCO_2}$  the mean measurement uncertainty of the nfCO<sub>2</sub> fluxes. The p-values describe the probabilities of observing the measured mean flux under the assumption (null hypothesis) that  $\bar{F}_{\rm nfCO_2} = 0$  and that deviations are solely due to measurement uncertainty. In addition, the minimum deviation from  $F_{\rm nfCO_2} = 0$  required to reject the null hypothesis at a significance level of 0.05 (minimum effect) and the number of samples required to detect a deviation from the null hypothesis of 3  $\mu$ mol m<sup>-2</sup> s<sup>-1</sup> at significance level of 0.05 and a power of 80 % ( $N_3$ ) is given.

| Variable                                                                                      | Zurich    |             | Paris         |               | Munich        |             |
|-----------------------------------------------------------------------------------------------|-----------|-------------|---------------|---------------|---------------|-------------|
|                                                                                               | Summer    | Winter      | Summer        | Winter        | Summer        | Winter      |
| $\overline{N}$                                                                                | 3         | 16          | 8             | 23            | 33            | 31          |
| $\bar{F}_{\mathrm{nfCO}_2}$ [ $\mu\mathrm{mol}\mathrm{m}^{-2}\mathrm{s}^{-1}$ ]               | $0 \pm 4$ | $1.5\pm2.7$ | $9.7 \pm 2.2$ | $2.7 \pm 2.1$ | $7.8 \pm 1.0$ | $5.3\pm1.1$ |
| $\overline{\delta F}_{\mathrm{nfCO}_2}  [\mu \mathrm{mol}  \mathrm{m}^{-2}  \mathrm{s}^{-1}]$ | 8.3       | 13.4        | 9.7           | 12.2          | 6.5           | 7.6         |
| p-value                                                                                       | 1.0       | 0.6         | < 0.001       | 0.2           | < 0.001       | < 0.001     |
| Minimum effect [ $\mu$ mol m <sup>-2</sup> s <sup>-1</sup> ]                                  | 8.3       | 5.3         | 4.3           | 4.1           | 1.9           | 2.3         |
| $N_3$                                                                                         | 61        | 158         | 83            | 129           | 37            | 50          |

The results show that for the Paris summer measurements and for the Munich measurements the mean ffCO<sub>2</sub>/CO<sub>2</sub> flux ratios were significantly different from 100 % (p-values < 0.05), with about 20 % non-fossil contribution in winter and 50 % (Munich) and 100 % (Paris) non-fossil contribution in summer. The small fossil component in Paris is surprising and not yet fully understood. In Zurich, no significant average nfCO<sub>2</sub> component was observed. While the small number of samples and the large measurement uncertainties in Zurich and Paris required a minimum non-fossil contribution of more than 20 % in winter and more than 40 % in summer to reject the null hypothesis / more than 300 measurements to detect a mean nfCO<sub>2</sub> contribution of 10 % at a power of 80 %, the minimum effect was reduced to 8 % in Munich and the required number of samples to about 90 summer measurements and 50 winter measurements, respectively. Similarly, the mean nfCO<sub>2</sub> fluxes were significantly different from zero for the Paris summer samples and the Munich samples. With the current setup, i.e., as in Munich, mean nfCO<sub>2</sub> fluxes of 3 µmol m<sup>-2</sup> s<sup>-1</sup> can be determined with about 40 to 50 measurements, which is close to the number of samples collected in this study.

Author contributions. LBo, ME, RK, and VL designed and built the REA flask sampler, and helped with its installation, maintenance, and setup improvements during the three measurement campaigns. AC wrote the logger programs. AC and SH acquired funding and managed the project. LE, BL, MR, JC, CH, and MM managed the installations of the three measurement sites. SS, LBi, and CL oversaw the IRGASON measurements in Zurich, Paris, and Munich, respectively, and helped, e.g., with updates of the logger program. RH operated the MGA<sup>7</sup> in all three cities, processed the EC data for the REA sample selection, and provided the final EC flux data used in this study. PA provided data from the midcost-sensors and helped with the installation in Munich. XG and JDC were responsible for the flask measurements at the ICOS Flask

785

790

and Calibration Laboratory and the ICOS Central Radiocarbon Laboratory, AJ and SP for the corresponding data processing and quality control. BM produced the footprint and surface-cover analysis based on the model of NK. AK operated the REA system, performed the analysis, and wrote the manuscript, with conceptual and methodological input and supervision from AC, SH, and NK. All authors reviewed and contributed to the manuscript.

Competing interests. The authors declare that they have no conflict of interest.

Acknowledgements. The authors have received funding from ICOS Cities, a.k.a. the Pilot Applications in Urban Landscapes – Towards integrated city observatories for greenhouse gases (PAUL) project, from the European Union's Horizon 2020 research and innovation program under grant agreement no. 101037319. Additional support was provided by internal funds and staff at the Universities of Heidelberg, Freiburg, and the Max Planck Institute for Biogeochemistry in Jena. Financial support from ICOS Switzerland (ICOS-CH) Phase 3 and Phase 4 (Swiss National Science Foundation, grants 20FI20\_198227, 20FI-0\_229655) is also acknowledged. We thank the following people for their contributions to this work: Felix Baab and Dirk Redepenning (University of Freiburg, Germany) for building the hardware at the REA inlet and logistics; Roland Vogt (University of Basel, Switzerland) and Carsten Jahn (KIT, Germany) for negotiations and installations at the Zurich and Munich sites; Pascal Rubli and Andrea Fischer (EMPA, Switzerland), Sophie Emberger (ETHZ, Switzerland), Sophie Bevini, Laura Bouillon, Ingrid Chanca, Lorna Foliot, Cécile Gaudry, and Guillaume Nief (LSCE, France), and Christian Becker and Klaus Kürzinger (TUM, Germany) for regular maintenance and logistics related to the REA flask sampler in Zurich, Paris, and Munich, respectively; Steffen Knabe and the entire staff of the ICOS Flask and Calibration Laboratory in Jena and the ICOS Central Radiocarbon Laboratory in Heidelberg for measuring the test and REA flasks; Hannes Juchem (University of Heidelberg) for providing CO<sub>2</sub> background concentration data from Mace Head; Matthias Zeeman (University of Freiburg, Germany) for managing the data infrastructure. DeepL was used for grammar and spell checking.

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
