# Peer review of "14C-based separation of fossil and non-fossil CO2 fluxes in cities using relaxed eddy accumulation: results from tall-tower measurements in Zurich, Paris, and Munich"

_EGUsphere, 2025_

## Referee Comment (RC1)

This paper evaluated the contribution of fossil fuel $CO_2$ (ff$CO_2$) emissions in Zurich, Paris, and Munich using both the relaxed eddy accumulation (REA) method and the eddy covariance (EC) method. The contribution of winter ff$CO_2$ was found to be about 80% in all three cities. For the summer, the measurement results in Munich demonstrated the major role of respiration, biofuels, and industrial processes that led to predominantly positive non-fossil $CO_2$ (nf$CO_2$) fluxes. The concentration of absolute $CO_2$ and $14CO_2$ of the REA samples was used to estimate the impact of ff$CO_2$ on local $CO_2$ concentration, namely, excess concentrations. In the end, the authors stated that this combination of REA and EC method can be useful for quantifying ff$CO_2$ and nf$CO_2$ fluxes, but challenges existed due to the limits of analytical capabilities. The topic is important given the current strategy to reduce carbon emissions, especially fossil fuel consumption. The idea of combining REA and EC to separate ff$CO_2$ and nf$CO_2$ fluxes is novel and could significantly improve the understanding of urban carbon sources. However, the experimental setup is questionable and lacks important descriptions. For example, the instrument calibration should be included in the method section, and the application of long intake lines to deliver air samples for REA flask sampling and MGA measuring should be further evaluated. Below are more comments in detail.

1, Please double-check the references. Some references need to be updated, such as Hilland et al. (2025) and Kunz et al. (2025).

2, Line 70-75. There are several timing issues that need to be clarified. Please describe the method applied to upsample 10 Hz MGA measurements to a 20 Hz dataset. Clock drift is a common issue in certain high-frequency instruments, and it appears that the MGA experienced this problem. Therefore, could you provide more information that shows your linear interpolation works great to solve this problem? How and how often did you check and sync the timestamps on the IRGASON and the MGA instrument?

3, Line 74. The correlation coefficient threshold (0.5) for determining poor correlation between the $CO_2$ time series seems too low. Were the IRGASON and MGA instruments calibrated using a standard $CO_2$ gas during the measurement? If calibration was conducted periodically, the $CO_2$ measured by IRGASON and MGA should be in much better agreement.

4, Line 93-94. Without the information about instrument calibration, the better agreement between MGA and flask measurements does not necessarily indicate that their $CO_2$ concentrations were accurate. Could you explain why the $CO_2$ concentration measured by IRGASON did not agree with the data measured by the other two instruments?

5, Line 109-110. I can see that the deadband is quite important parameter in the REA method. Since the dynamic deadbank was more suitable, it would be great for other researchers interested in this method if you could provide more details on this method, such as how to determine the scaling factor $\delta$.

6, Line 114-116. It looks like the REA flask samples were analyzed by different laboratories. Please describe how consistency in flask sample results between laboratories was ensured?

7, Line 120. Eventually, the β was determined based on the co-located high-frequency EC data; The statement in Lines 120–121 may cause confusion. I recommend either removing this sentence or comparing this 0.627 with the β calculated based on EC data, and showing the advantage of your method.

8, Line 130. Please elaborate on the limitations and uncertainties introduced by assuming scalar similarity between CO2 and 14CO2 and using their ratio to derive ffCO2 fluxes?

9, Line 150-151. Since the storage flux correction is not feasible for REA measurement. What did the storage term look like compared with your EC results? Additional discussion may be helpful to demonstrate the minimal impact of storage flux correction.

10, Line 214-215. I can see that the inlet of the MGA instrument was located close to the flask sample inlet, so most likely, the intake line of the MGA was the same setup as the flask sampling. If so, please clarify in the manuscript. My question is, do both intake lines always share the same flow rate? Given the 100 m intake lines in Munich (assuming the same intake line setup), pressure drop could be an issue for the MGA instrument. Is there evidence that shows that the MGA instrument provides solid readings with low inlet pressure? Given the fact that MGA results were in better agreement with flask results than IRGASON, the quality check of the MGA instrument is important.

11, Line 243. The flux sites were located on the edge of a telecommunications tower. Please discuss the impact of the tower structure on the micrometeorological environment.

12, Line 270 and Table 2. Please add the flow rate of the REA intake lines and the corresponding Reynolds number to show that turbulent flow through the lines was well-maintained. The flow rates of 7 L/min in Paris and 11 L/min in Munich may result in laminar flow in the intake lines. Please add discussions about the potential impact. The information about the setup for MGA sampling should also be included, as well as the lagtime determined.

13, Line 345. I totally agree with the authors that great uncertainty exists in determining the storage term, especially for urban flux measurement. In the manuscript, storage was calculated using the single-point profile. It would be great to conduct additional storage calculations using the CO2 concentration measured at two heights in Munich and compare with the single-point method. The authors set a threshold of 20 $\mu$mol m$^{-2}$ s$^{-1}$ to flag extremely large storage terms. Could you clarify the criterion used to determine this threshold?

14, Line 365-368. Obviously, there were many fewer negative ffCO2 flux data points in Munich than other two cities. It may be worth noting this to highlight the advantages of the REA measurements in Munich.

15, Line 385-391. Based on the data measured in Zurich and Paris, the ffCO2 fluxes in winter were systematically higher than in summer, while it looks like there was no significant difference between

summer ffCO2 fluxes and winter ffCO2 fluxes in Munich. Could you add some discussion? Probably the contribution of natural gas consumption?

16, Line 491-492. It would be great to provide the exact proportions of the vegetated area relative to the footprint areas and have this discussed in the nfCO2 fluxes comparison between cities.

17, Line 492-495. Do the 2.5 $\mu$mol m$^{-2}$ s$^{-1}$ values represent the mean annual human respiration fluxes for one city, or are they the same for all three cities? Please clarify.

18, Line 727-729. Although the authors acknowledge the high-frequency attenuation caused by the MGA intake lines, no spectral analysis was provided to evaluate this effect, nor were potential factors affecting the coverage of energy-carrying eddies discussed. Please add more related information.